# Hierarchical Implicit Neural Emulators

**Ruoxi Jiang**[1,5]   **Xiao Zhang**[1]   **Karan Jakhar**[3,6]   **Peter Y. Lu**[4,7]
**Pedram Hassanzadeh**[3]   **Michael Maire**[1]   **Rebecca Willett**[1,2]

[1]Department of Computer Science, The University of Chicago
[2]Department of Statistics, The University of Chicago
[3]Department of Geophysical Sciences, The University of Chicago
[4]Department of Physics, The University of Chicago
[5]Artificial Intelligence Innovation and Incubation Institute, Fudan University
[6]Department of Mechanical Engineering, Rice University
[7]Department of Electrical and Computer Engineering, Tufts University

## Abstract

Neural PDE solvers offer a powerful tool for modeling complex dynamical systems, but often struggle with error accumulation over long time horizons and maintaining stability and physical consistency. We introduce a multiscale implicit neural emulator that enhances long-term prediction accuracy by conditioning on a hierarchy of lower-dimensional future state representations. Inspired by the stability properties of numerical implicit time-stepping methods, we developed an approach that leverages predictions several steps ahead in time at increasing compression rates for next-timestep refinements. By actively adjusting the temporal downsampling ratios, our design enables the model to capture dynamics across multiple granularities and enforce long-range temporal coherence. Experiments on turbulent fluid dynamics show that our method achieves high short-term accuracy and produces long-term stable forecasts, significantly outperforming non-hierarchical autoregressive baselines while adding minimal computational overhead. The codebase is available at this link[1].

## 1  Introduction

Machine learning (ML)-based surrogate modeling of dynamical systems has spurred great interest in recent years due to transformative applications in climate modeling [1–5], molecular dynamics [6–8], cosmology [9–11], and beyond. These surrogate models or emulators are often orders of magnitude faster than classical numerical solvers and can be trained to directly emulate real-world data. Although existing developments in neural surrogate models have made notable strides in short-term prediction, often benchmarked using one-step mean square error (MSE), their ability to deliver stable long-term predictions remains a critical challenge [12–14]. For scientific practitioners, achieving a long-term stable forecast is valuable and expands the potential for research use, especially where classical numerical solvers are too costly to run. For example, climate scientists can use accurate long-term emulators to predict the probability of extreme events [15]. However, the error accumulation in unrolling the predictions for complex, nonlinear, and multiscale dynamical systems often causes existing surrogate models to degrade catastrophically during long-term forecasting, resulting in unphysical collapsed or exploding solutions [14, 16].

Recent efforts to address long-term stability explore two primary avenues. From the temporal evolution perspective, frameworks centered on multi-step ahead prediction aim to shift focus from single-step accuracy to extended horizons by training models on multi-step targets [17–20]. Yet,

---

*Correspondence to: roxie_jiang@fudan.edu.cn, zhang7@uchicago.edu, and willett@uchicago.edu.

[1]https://github.com/roxie62/Hierarchical-Implicit-Neural-Emulators

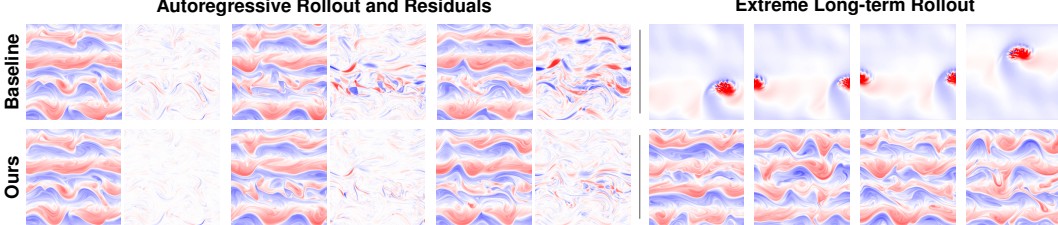

Figure 1: In a chaotic, turbulent system, our emulator achieves accurate short-to-mid-term rollout predictions (*left*). Even over extremely long sequences (up to $10^5$ emulation steps, *right*), it captures the physical jet patterns, while baseline autoregressive methods quickly drift and break down.

such methods can underperform even one-step ahead autoregressive baselines for short-term complex dynamics [21]. From the spatial domain perspective, architectures that use frequency decompositions or long-term statistics as training targets advance the performance of surrogate modeling to better handle the multiscale structure of the data [22–26]. However, they often face a trade-off between short-term accuracy and long-term robustness, or require additional physical knowledge.

In this work, we draw inspiration from numerical analysis and reframe neural surrogate modeling through the lens of time-stepping methods for solving differential equations. Viewing the conventional autoregressive models from this perspective, we can interpret their failure patterns as analogous to the instability of explicit schemes (e.g., forward Euler), especially for larger timesteps and in stiff systems. To improve stability (Fig. 1), we propose to enhance the autoregressive system based on an analogy with the implicit time-stepping methods, which are known for their unconditional stability in challenging integration scenarios [27].

The cornerstone of our approach is the use of *future* frame information to guide next-step predictions, fundamentally departing from existing autoregressive frameworks constrained to the use of past temporal states. While inspired by numerical implicit schemes that use iterative solvers to refine predictions, our framework avoids the corresponding computational overhead by effectively performing the iterative refinement steps in parallel. This is achieved by simultaneously forecasting a hierarchy of future states during autoregressive rollout (Fig. 2), resulting in nearly the same number of forward passes compared to the one-step ahead baseline model. Furthermore, unlike prior multi-step ahead approaches [17–19], our framework actively compresses future states into a hierarchical representation, balancing accuracy and computational tractability while enabling feedback from future time frames. On 2D turbulence with multiscale and chaotic structure, our method demonstrates greatly enhanced long-term stability compared to the baseline models (Fig. 1, *right*), while introducing a minimal amount of additional computational overhead.

We summarize our contributions as follows:

- **Neural emulation with an implicit schema and a hierarchical structure.** We propose a novel emulator that uses an augmented state built from a hierarchy of coarse-grained representations of future states. This approach captures multiscale interactions and — by autoregressively applying the emulator to the augmented state — efficiently mimics the iterative refinement mechanism of implicit solvers, achieving better accuracy and stability.

- **Accurate simulation of chaotic systems.** Our approach is well-suited for turbulent flows, which are chaotic and have complex multiscale structures. On a 2D Navier-Stokes system with Reynolds number of $10^4$, our model achieves substantial improvements — reducing mean squared error by over 50% for predictions 25 to 50 steps, compared to the standard 1-step ahead baseline.

- **Stability for extremely long rollouts that are $10\times$ the training length.** Our emulator exhibits strong stability over extremely long sequences. For the 2D Navier-Stokes system at $10^4$ Reynolds number, it rolls out up to $\mathbf{2 \times 10^4}$ steps (one time the length of the training set) without empirical failure. Even at $\mathbf{2 \times 10^5}$ steps (ten times the length of the training set), it maintains a low failure rate of just 7.0%, far outperforming existing baselines, which typically fail much earlier and for all initial conditions.

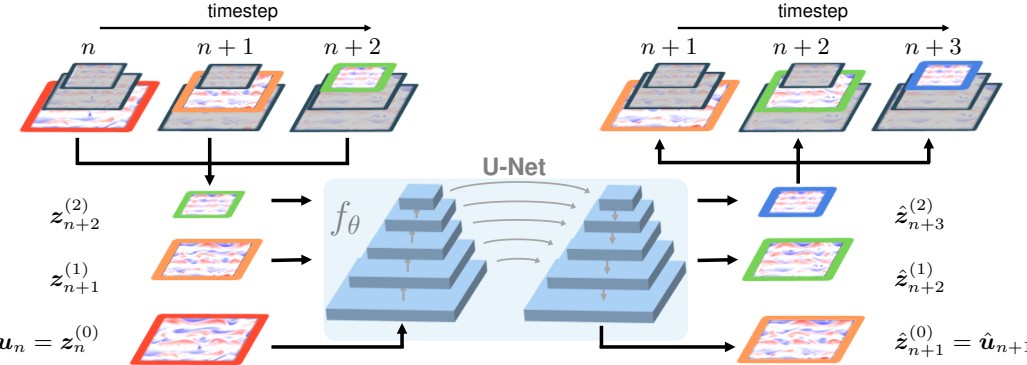

Figure 2: **Diagram of our hierarchical implicit neural emulator.** Our model approximates the dynamics $u$ using an implicit formulation with hierarchical latent representations. For each trajectory $u_n$, we augment its state with multiscale latent features $z_n^{(l)}$, obtained by downsampling $u_n$. The model predicts the future state $\hat{u}_{n+1}$ and long-horizon latent representations $\hat{z}_{n+2}^{(1)}, \hat{z}_{n+3}^{(2)}$, enabling it to capture long-term dynamics. Following the implicit schema, the model conditions on both the past trajectory $u_n$ and future latent variables $z_{n+1}^{(1)}, z_{n+2}^{(2)}$ during training, while using predictions of future states computed in the previous step of an autoregressive rollout during inference, providing richer context to effectively mitigate error accumulation for long-term predictions.

## 2 Related Work

**Hierarchical networks.** Many physical systems exhibit patterns across spatial and temporal scales. To effectively emulate such dynamical systems, the model architecture should reflect their inherent hierarchical structure. A U-Net [28], for instance, introduces skip connections between different spatial resolutions, enabling effective reasoning across both high-level semantics and fine-grained details. Many subsequent approaches similarly integrate hierarchical structure into a system's data representations or computational elements. Feature Pyramid Networks [29] fuse multiscale features to improve prediction accuracy. Multiscale Vision Transformers [30] construct hierarchical representations tailored for attention mechanisms. Inspired by multigrid solvers [31], multigrid CNNs [32, 33] maintain multiscale representations at each layer and introduce cross-scale convolution operations to enable efficient information routing via bidirectional connections between adjacent scales. Hierarchical VAEs (HVAEs) [34] extend latent variables across multiple scales, significantly improving generative model quality. However, HVAEs often face challenges such as high training variance and instability. Nested Diffusion Models [35] address similar challenges by pretraining and freezing the hierarchical latents, then training a collection of conditional diffusion models to generate each level of the latent hierarchy conditioned on those above, concluding with image generation.

**Neural PDE solvers.** Neural PDE solvers offer a data-driven alternative to traditional numerical methods, with promising applications in real-world applications, such as weather forecasting (e.g., FourCastNet [1], GenCast [36]). These methods fall into two main classes: (1) neural operators, which map between function spaces (e.g., FNO [22], DeepONet [37]), and (2) grid-based architectures, which operate on a discretized grid or mesh (e.g., U-Nets [28]). While neural operators promise mesh-agnostic inference, grid-based methods remain prevalent due to their simplicity and popularity in computer vision. Although our current implementation assumes grid-structured inputs, the framework can be naturally extended to mesh-based representations, as demonstrated in prior work [38–40].

A major challenge for both is long-term stability, where small errors amplify over time. Stabilization efforts include: (1) data-driven methods like pushforward training [18] and PDE-Refiner [41], which improve rollouts but suffer from instability and high computational cost; (2) physics-informed strategies such as multi-time stepping [42], which embed physical priors but limit data-driven flexibility; and (3) architectural advances like FNOs [22] and Wavelet Neural Operators [23], which improve spatial modeling but struggle with temporal coherence and optimization. Despite progress, current approaches fall short in handling multiscale chaotic systems, such as turbulent flows, where spatiotemporal complexity demands both efficient learning and robust long-horizon predictions.

# 3 Problem Formulation

**Dynamical systems.** Consider a nonlinear dynamical system evolving over time as:

$$\frac{\partial u}{\partial t} = f(u, x, t, \nabla_x u, \nabla_x^2 u \dots), \quad u(x, 0) = u_0, \tag{1}$$

where $f$ encodes the unknown governing physics. Let $u(x, t) \in \mathbb{R}$ denote the continuous trajectory of dynamics in function space $\mathcal{U}$, and $\boldsymbol{u}_t \in \mathbb{R}^m$ represent discretized snapshots (i.e., sample from a finite spatial grid with $m$ points) at time $t$. To numerically approximate the dynamics, we adopt discrete methods (e.g., finite differences, spectral methods), transforming the PDE into a system of ODEs:

$$\frac{d\boldsymbol{u}_t}{dt} = \tilde{f}(\boldsymbol{u}_t, t), \quad \boldsymbol{u}_0 \in \mathbb{R}^m, \tag{2}$$

where $\tilde{f}$ corresponds to the temporal dynamics at discretized states. Given $N + 1$ consecutive observations $\{\boldsymbol{u}_0, \boldsymbol{u}_1, \dots, \boldsymbol{u}_N\}$, *our goal* is to learn a neural emulator $f_\theta$ that approximates the underlying dynamics and predicts future states. The emulator is parameterized by $\theta$.

**Autoregressive models.** Autoregressive methods learn a transition operator $f_\theta : \mathbb{R}^m \to \mathbb{R}^m$ to iteratively predict next state $\hat{\boldsymbol{u}}_{n+1}$ from $\boldsymbol{u}_n$:

$$\hat{\boldsymbol{u}}_{n+1} = f_\theta(\boldsymbol{u}_n), \tag{3}$$

and chain predictions via recursive rollouts:

$$\hat{\boldsymbol{u}}_{n+k} = f_\theta(f_\theta(f_\theta \dots (\boldsymbol{u}_n))) \quad \text{for } k \text{ steps.} \tag{4}$$

While efficient and accurate for short-term predictions, autoregressive methods often suffer from error accumulation over long horizons, where small discrepancies compound over time [18, 14]. Our approach adheres to this autoregressive framework, but introduces mechanisms to mitigate these long-term errors, as described in Section 4.

# 4 Method

In this section, we show how to improve short-term forecast and preserve long-term coherence in an autoregressive model by building an implicit multiscale framework.

## 4.1 Implicit Neural Emulator

**Explicit *v.s.* implicit methods.** For the semi-discretized system in Eqn. 2, explicit time-stepping methods [43] compute future states directly through:

$$\boldsymbol{u}_{n+1} \approx F(\boldsymbol{u}_n), \tag{5}$$

where the mapping $F(\cdot)$ represents an explicit update rule depending only on the current state $\boldsymbol{u}_n$. For instance, in the *forward* Euler method [44], it takes the form:

$$F(\boldsymbol{u}_n) := \boldsymbol{u}_n + \Delta t \tilde{f}(\boldsymbol{u}_n) \tag{6}$$

to compute an estimate of $\boldsymbol{u}_{n+1}$. This approach can be connected to the autoregressive learning seen in Eqn. 3, where predictions rely solely on past states. And similar to autoregressive methods, it is known to suffer from the instability using a larger time step $\Delta t$ or facing systems with stiffness [45], where adaptive and higher-order variants are usually required to ensure stability of the solver, which substantially increases the computational cost.

Implicit methods are designed to solve these issues: they leverage both current states $\boldsymbol{u}_n$ and future states $\boldsymbol{u}_{n+1}$ to solve the following implicit equation:

$$\boldsymbol{u}_{n+1} = F(\boldsymbol{u}_n, \boldsymbol{u}_{n+1}). \tag{7}$$

Here $F(\boldsymbol{u}_n, \boldsymbol{u}_{n+1}) := \boldsymbol{u}_n + \Delta t \tilde{f}(\boldsymbol{u}_{n+1})$ in the *backward* Euler method. Solving $\boldsymbol{u}_{n+1}$ in Eqn. 7 requires extra computations to find the root. A classic example is Newton's method, which iteratively updates the estimate via:

$$\boldsymbol{u}_{n+1}^{i+1} = \boldsymbol{u}_{n+1}^i - \frac{F(\boldsymbol{u}_n, \boldsymbol{u}_{n+1}^i)}{F'(\boldsymbol{u}_n, \boldsymbol{u}_{n+1}^i)} \tag{8}$$

where $i$ indicates the iteration index. Compared to explicit methods, implicit approaches can accommodate much larger timesteps while maintaining stability, offering computational efficiency for long-time integration.

**Implicit neural emulator with two-step prediction.** Drawing inspiration from the robustness of implicit solvers, we propose a neural network architecture that combines the stability of implicit methods with the simplicity of explicit methods. Our goal is to address the compounding error typically seen in autoregressive models by introducing a structure that implicitly reasons about future states. Rather than solving the implicit equation in Eqn. 7 directly — which would entail iterative root-solving at each step — we adopt a relaxed version. We introduce a latent variable $z_{n+1} = T(u_{n+1})$, which represents an abstract encoding of the future state $u_{n+1}$. This latent variable can be directly computed during training, but must be predicted at test time, leading to the following formulation:

$$\hat{u}_{n+1}, \hat{z}_{n+2} = f_\theta\left(u_n, z_{n+1}\right), \tag{9}$$

where the network is trained to simultaneously predict the next physical state $\hat{u}_{n+1}$ and the latent representation $\hat{z}_{n+2}$ that encodes information about a future state.

This architecture mirrors the iterative refinement process used in implicit solvers. Our model is given an initial approximation of state $n + 1$ represented by $z_{n+1}$ and then refines this approximation by applying $f_\theta(u_n, z_{n+1})$ to produce the predicted physical state $\hat{u}_{n+1}$. By also providing $\hat{z}_{n+2}$ as an auxiliary output, the model is able to perform this refinement process as part of the standard autoregressive rollout across timesteps, effectively treating $(u_n, z_{n+1})$ as an augmented state variable.

In practice, we choose the transformation $T$ to be a downsampling or coarse-graining operation. Predicting a coarse-grained $\hat{z}_{n+2}$ rather than the full physical state $\hat{u}_{n+2}$ encourages the emulator to first capture large-scale spatial features and, as shown in our experiments, leads to better prediction accuracy. We will elaborate on the design of the transformation $T$ in a later subsection.

## 4.2 Hierarchical Multi-step Prediction

The architecture above naturally extends to a hierarchical multi-step modeling framework in which predictions are conditioned on multiple latent representations of anticipated future states. We denote these representations as $z_m^{(l)} = T^{(l)}(u_m)$, where $l$ indexes increasing levels of abstraction and we assume $u_m = z_m^{(0)}$. The model is then trained to predict across $L$ hierarchical levels as follows:

$$\hat{u}_{n+1}, \hat{z}_{n+2}^{(1)}, \ldots, \hat{z}_{n+L}^{(L-1)} = f_\theta\left(u_n, z_{n+1}^{(1)}, \ldots, z_{n+L-1}^{(L-1)}\right). \tag{10}$$

As shown in Figure 2, our model effectively operates in an augmented state space formed by the concatenation of the current state with future latent states of increasing levels of abstraction. In practice, this simple strategy proves scalable and robust, and easily extensible to deeper hierarchies.

This generalized framework enhances the analogy to the multi-step iterative refinement used in implicit solvers while also enabling the model to capture broader temporal dynamics, which helps mitigate error accumulation by informing each prediction with a richer view of future latent dynamics.

**Leveraging future context in a hierarchical fashion.** Using future context offers two complementary benefits. From an *encoding* standpoint, it allows the model to process input as a structured sequence where immediate states like $u_n$ retain fine-grained detail, while distant latents such as $z_{n+l}^{(l)}$ provide coarse-scale insights like aggregate dynamics or long-term trends. This mirrors the philosophy of multi-step implicit methods like Adams–Moulton [46], which integrate information from multiple past states to produce more accurate future estimates, though with a slight distinction that we are conditioning on the future frames, while being able to extend to include the history states as well.

From a *decoding* standpoint, learning to predict distant states becomes progressively harder due to increased uncertainty and error amplification. By supervising the model across multiple levels of abstraction, we encourage a balanced learning process where both local precision and global structure are prioritized. Viewed through the lens of implicit solvers, this can be interpreted as executing $L$ steps of iterative refinement, distributed across temporal frames and abstraction levels.

## 4.3 Training Details

**Choosing abstract transformations.** A key component of training our model is the choice of the transformations $T^{(l)}$, which map physical states $u_n$ to latent representations $z_n$. There are

| | Confi. Name | Model | Config. Name | Model |
|---|---|---|---|---|
| Ours | $L = 2$ | $\hat{\boldsymbol{u}}_{n+1}, \hat{\boldsymbol{z}}_{n+2}^{(1)} = f_\theta(\boldsymbol{u}_n, \boldsymbol{z}_{n+1}^{(1)})$ | $L = 3$ | $\hat{\boldsymbol{u}}_{n+1}, \hat{\boldsymbol{z}}_{n+2}^{(1)}, \hat{\boldsymbol{z}}_{n+3}^{(2)} = f_\theta(\boldsymbol{u}_n, \boldsymbol{z}_{n+1}^{(1)}, \boldsymbol{z}_{n+2}^{(2)})$ |
| Baseline | $L = 1$ | $\hat{\boldsymbol{u}}_{n+1} = f_\theta(\boldsymbol{u}_n)$ | Spatial Hierarchy | $\hat{\boldsymbol{u}}_{n+1}, \hat{\boldsymbol{z}}_{n+1}^{(1)}, \hat{\boldsymbol{z}}_{n+1}^{(2)} = f_\theta(\boldsymbol{u}_n, \boldsymbol{z}_n^{(1)}, \boldsymbol{z}_n^{(2)})$ |
| | 2-Step Ahead | $\hat{\boldsymbol{u}}_{n+1}, \hat{\boldsymbol{u}}_{n+2} = f_\theta(\boldsymbol{u}_n)$ | History Hierarchy | $\hat{\boldsymbol{u}}_{n+1} = f_\theta(\boldsymbol{u}_n, \boldsymbol{z}_{n-1}^{(1)}, \boldsymbol{z}_{n-2}^{(2)})$ |
| | 2-Step History [19] | $\hat{\boldsymbol{u}}_{n+1} = f_\theta(\boldsymbol{u}_{n-1}, \boldsymbol{u}_n)$ | 3-Step History [19] | $\hat{\boldsymbol{u}}_{n+1} = f_\theta(\boldsymbol{u}_{n-2}, \boldsymbol{u}_{n-1}, \boldsymbol{u}_n)$ |

Table 1: **Model and baseline configurations.** We conduct experiments with our model designs ($L = 2, L = 3$) and several baseline configurations for comprehensive analysis.

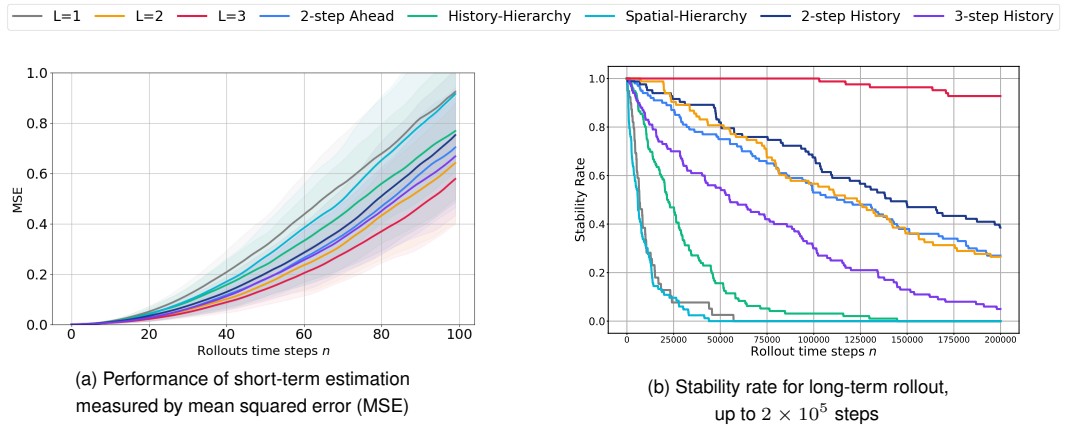

(a) Performance of short-term estimation measured by mean squared error (MSE)

(b) Stability rate for long-term rollout, up to $2 \times 10^5$ steps

Figure 3: **Short-term accuracy vs. long-term robustness.** *Left:* MSE trend over a 100-step autoregressive rollout. Solid lines indicate the average performance, and shaded regions represent one standard deviation. *Right:* We evaluate the stability rate over 10 times the training sequence length across 100 trials with various initial conditions. Stability is defined by maintaining physical statistics (conserved kinetic energy). Our method with $L = 3$ achieves a 93% stability rate, significantly outperforming all baselines, with the next best, 2-Step History, at just 38%, indicating superior short-term accuracy and long-term robustness.

several viable options for $T^{(l)}$, such as a learnable encoder or a predefined information-reduction function (e.g., spatial pooling). While a learned encoder offers expressiveness, it can introduce instability during training due to the constantly shifting latent representations. Instead, we focus on the multiscale structure of spatiotemporal systems and choose $T^{(l)}$ to be a set of fixed spatial downsampling transformations:

$$\boldsymbol{z}_n^{(l)} = T^{(l)}(\boldsymbol{u}_n) := \text{DownSample}(\boldsymbol{u}_n, r^l), \tag{11}$$

Here $r^l$ denotes the predefined downsampling rate for the $l$-th level with $r^l \leq r^{l+1}$, providing a sequence of increasingly coarse-grained states.

**Training objective.** Finally, our training loss is designed to supervise both the predicted physical state and the associated abstract latents:

$$\ell(\theta) = d(\hat{\boldsymbol{u}}_{n+1}(\theta), \boldsymbol{u}_{n+1}) + \sum_{l=1}^{L-1} d(\hat{\boldsymbol{z}}_{n+l}^{(l)}(\theta), \boldsymbol{z}_{n+l}^{(l)}), \tag{12}$$

where $d(\cdot)$ denotes a distance metric such as $l1$ or $l2$ loss for simplicity, though this could be replaced with domain-specific alternatives. This objective ensures that the model's predictions remain aligned with both immediate and long-term dynamics, across all abstraction levels.

# 5  Experiments

In this section, we benchmark our implicit modeling framework for 2D turbulence prediction. This canonical flow has been extensively used for testing novel ML-based schemes [47–49, 41, 50].

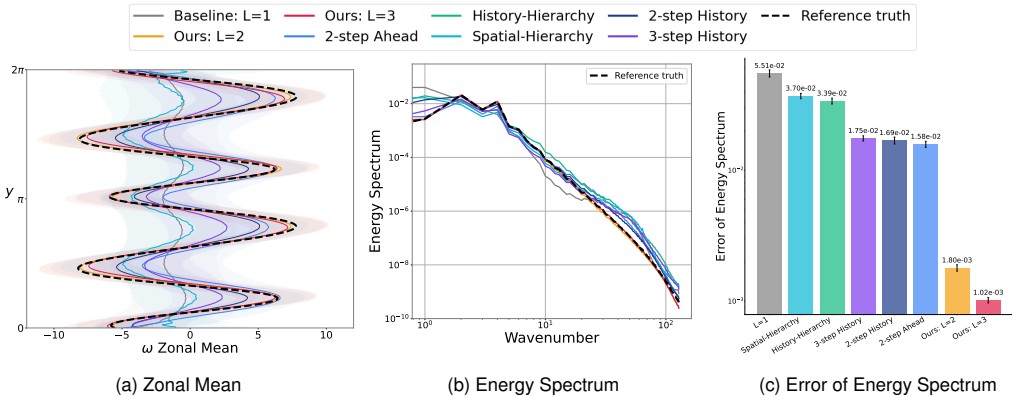

(a) Zonal Mean        (b) Energy Spectrum        (c) Error of Energy Spectrum

Figure 4: **Further investigation of long-term stability of** $2 \times 10^5$ **steps (10× the training length) rollout.** *(a):* Time-averaged zonal mean of vorticity comparing ground truth (dashed black lines) with emulator runs. Only our systems with $L = 2$ and $L = 3$ accurately follow the ground truth trend, with the reference truth lies well within error bars, demonstrating that our emulator accurately captures the system's mean statistics. In contrast, other methods significantly deviate, indicating a substantial drift and failure to preserve structure. *(b,c):* Spectrum of long-term rollout for normalized data.

**Navier-Stokes.** We focus on the dimensionless vorticity-streamfunction ($\omega - \psi$) formulation of the incompressible Navier-Stokes equations in a 2D $x - y$ domain [47, 48]:

$$\frac{\partial \omega}{\partial t} + \mathcal{N}(\omega, \psi) = \frac{1}{Re} \nabla^2 \omega - \chi \omega + f + \beta v_y, \tag{13}$$

where $\nabla^2 \psi = -\omega$, $\boldsymbol{v} = (v_x, v_y)$ is velocity, and $\omega = \nabla \times \boldsymbol{v}$. $\mathcal{N}(\omega, \psi)$ is the Jacobian and represents non-linear advection. The flow is defined by a Reynolds number $Re = 10^4$, time-constant forcing $f$ at a given wavenumber, and a Rayleigh drag $\chi = 0.1$. The Coriolis parameter, $\beta = 20$, induces zonal jets characteristic of geophysical turbulence, mimicking the influence of Earth's rotation on atmospheric and oceanic flows [51]. The domain is doubly periodic with length $L = 2\pi$. Training data is generated using "py2d" [52] on a $512 \times 512$ grid, with 18,000 $\omega$ snapshots. For analysis in this paper, the data is downsampled to a $256 \times 256$ grid, except in Section 5.3, which discusses ablation studies on different datasets and resolutions.

**Architecture.** We adopt the UNet design following Dhariwal and Nichol [53] as our base model, following the standard setup with using a structured dataset. Specifically, our model has an encoder composed of 5 downsampling blocks that reduce the input to an $8 \times 8 \times 256$ feature map at the bottleneck. The decoder mirrors the encoder structure to reconstruct the output. To improve the model's ability to capture high-frequency details, we incorporate Fourier layers [22] into each block to better handle high-frequency information (see details in Appendix A.3).

Given that our design must accommodate images at multiple resolutions, we align with the UNet's hierarchical information flow: lower-resolution inputs are introduced at the earlier encoder stages, while decoder blocks operating at matching resolutions are used to produce the final output. This structure ensures that the most abstract features are captured at the encoder's bottleneck, promoting a coherent and structured flow of information throughout the network.

**Model configurations and baseline setup.** Our experiments primarily focus on designs with two-level and three-level hierarchies, indicated by $L = 2$ and $L = 3$. For the input with $256 \times 256$ resolution, we choose $r^1 = 8$ and $r^2 = 32$ as our default downsampling rate for latent variables $\boldsymbol{z}_{n+1}^{(1)}, \boldsymbol{z}_{n+2}^{(2)}$ respectively. To rigorously assess our model's performance, we also propose several baseline methods that diverge from our formulation, including different setups of input/output hierarchy configurations, as detailed in Table 1.

**Hierarchical autoregressive rollout.** To enable autoregressive rollout given only the initial condition $\boldsymbol{u}_n$, we introduce specially-designed corner-case inputs, corresponding to $L - 1$ higher-level latent slots, during both training and inference. During the prediction, with a small probability $p$ (see details in Appendix A.3), we sample training instances in which the model receives a partially missing hierarchy of latent inputs and is tasked to reconstruct the missing parts. Concretely, for our $L = 2$

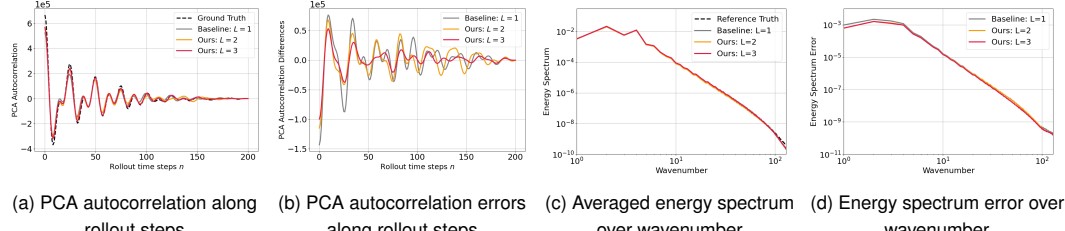

| (a) PCA autocorrelation along rollout steps | (b) PCA autocorrelation errors along rollout steps | (c) Averaged energy spectrum over wavenumber | (d) Energy spectrum error over wavenumber |

Figure 5: **Ablation studies on the design of the hierarchy.** We evaluate our system across different hierarchy levels $L$ and visualize the results using: *(a)* PCA autocorrelation and *(c)* energy spectrum averaging over 200 rollout steps, with corresponding deviations from the ground truth shown in *(b)* and *(d)*, respectively. Our hierarchical design with $L = 3$ outperforms the baseline $L = 1$, achieving better alignment with temporal structure and high-frequency components.

|  | 1 | 25 | 50 | 75 | 100 |
|---|---|---|---|---|---|
| $r^1 = 1$ | 9.900e-04 (2.143e-04) | 5.865e-02 (2.890e-02) | 2.103e-01 (8.888e-02) | 4.470e-01 (1.762e-01) | 7.738e-01 (2.520e-01) |
| $r^1 = 2$ | 1.747e-03 (3.942e-04) | 1.146e-01 (6.328e-02) | 3.774e-01 (1.797e-01) | 7.082e-01 (3.088e-01) | 1.023e+00 (3.352e-01) |
| $r^1 = 4$ | 7.735e-04 (1.954e-04) | 5.651e-02 (3.199e-02) | 1.998e-01 (1.074e-01) | 4.225e-01 (1.979e-01) | 7.280e-01 (2.522e-01) |
| $r^1 = 8$ | 5.248e-04 (1.148e-04) | **4.024e-02 (1.920e-02)** | **1.606e-01 (6.496e-02)** | **3.731e-01 (1.509e-01)** | **6.547e-01 (2.440e-01)** |
| $r^1 = 16$ | **5.155e-04 (1.138e-04)** | 4.898e-02 (2.011e-02) | 2.039e-01 (8.009e-02) | 4.416e-01 (1.789e-01) | 7.399e-01 (2.658e-01) |

Table 2: **Ablation study on downsampling ratio ($r^1$) for our $L = 2$ model on $256 \times 256$ resolution data with jet.** We evaluate model performance by computing roll-out mean squared error (MSE) for downsampling ratios $r^1 = 1, 2, 4, 8, 16$, and present the mean (standard deviation) over 100 trials with varied initial conditions. Notably, $r^1 = 8$ exhibits the minimal error, indicating an ideal balance between preserving contextual information and mitigating estimation challenges.

levels design, we present the model with $[\boldsymbol{u}_n, \boldsymbol{0}]$, where $\boldsymbol{0}$ denotes the zero-filled tensor matching the shape of the higher hierarchy latent $\hat{\boldsymbol{z}}^{(1)}$, and train it to predict $\hat{\boldsymbol{z}}^{(1)}_{n+1}$. This ensures that the model learns to infer higher-level latent states from just the initial condition. At the evaluation time, each method is provided only with the initial condition $\boldsymbol{u}_n$. In the hierarchy $L = 2$, we first input $[\boldsymbol{u}_n, \boldsymbol{0}]$ to obtain $\hat{\boldsymbol{z}}^{(1)}_{n+1}$. Then, using the full spatial-temporal hierarchy states $[\boldsymbol{u}_n, \hat{\boldsymbol{z}}^{(1)}_{n+1}]$, we continue with autoregressive rollout.

## 5.1 Short-term Accuracy and Long-term Stability

We assess our model's performance under two scenarios: (1) Short-term predictions, up to 100 steps using mean squared error (MSE); and (2) Extremely long-term predictions, generating up to $2 \times 10^5$ steps to evaluate the alignment of predicted and true physical statistics.

**Evaluation on short-term estimation.** Figure 3(a) presents MSE results comparing our method with various baselines. All methods show similar 1-step MSE, yet the 2-Step Ahead Baseline performs poorly (see Table 6 in Appendix A.1). Our models with $L = 3$ and $L = 2$ consistently provide accurate estimates, surpassing all baselines despite challenges in recursive prediction error accumulation. Baselines using multiple input or output frames (*e.g.*, 3-Step History or 2-Step Ahead) perform better for *mid-term* rollout, highlighting the importance of processing temporal sequences in autoregressive models. In comparison, basic hierarchical baselines (*e.g.*, History-Hierarchy or Spatial-Hierarchy) provide minimal gains, suggesting that temporal hierarchy is crucial.

**Evaluation on long-term estimation.** Over very long rollouts, mean squared error (MSE) becomes an unreliable metric for evaluating autoregressive systems. Instead, we verify if the predictions align with the true physical statistics, such as the system's energy given by $E = \frac{1}{2} \left( v_x^2 + v_y^2 \right)$. For a statistically stationary turbulent system, like the one studied here, this energy stays around a constant mean due to the balance between energy input by the deterministic forcing and its dissipation via viscosity and the linear drag in equation 13 [54].

**Stability rate.** We use a simple binary energy-based approach to check if a prediction adheres to correct physical patterns. By calculating the mean and standard deviation of energy from a 200K-step true trajectory, we ensure all true trajectory energy values fall within 4 standard deviations of the mean.

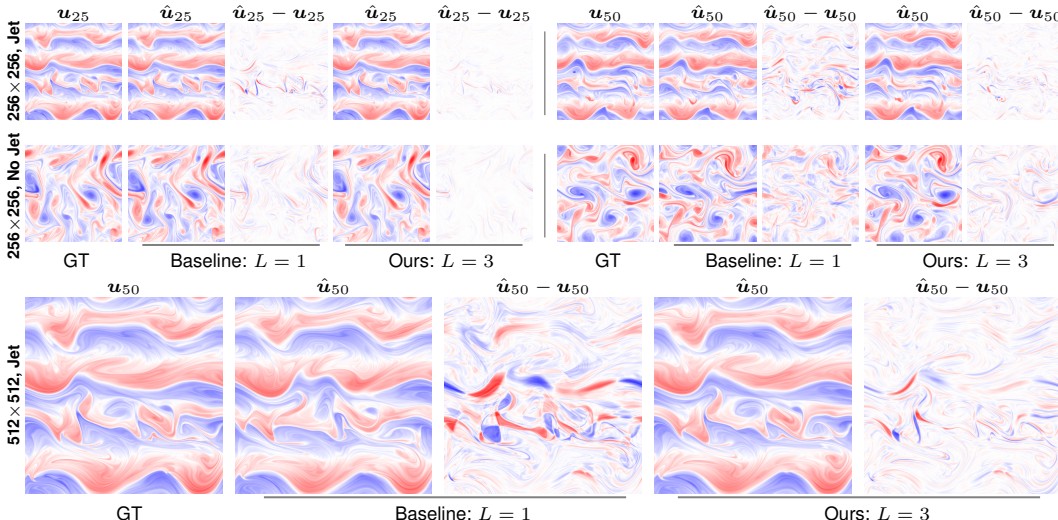

Figure 6: **Visualization of rollout estimation across multiple datasets.** We apply our approach to three different flows: (1) $Re = 10^4$, $256 \times 256$ resolution featuring zonal jets, (2) $Re = 5 \times 10^3$, $256 \times 256$ resolution without zonal jets, and (3) $Re = 10^4$, $512 \times 512$ resolution with zonal jets. Our method ($L = 3$) gives more accurate predictions with lower associated residuals in all three scenarios.

A prediction is deemed stable if its energy remains within 5 standard deviations of the true mean (see Appendix A.2). Appendix B.2 includes examples with varying energy deviations, illustrating that predictions exceeding this threshold exhibit incorrect physical patterns.

Figure 3(b) presents the stability rates for all methods, based on predictions over 200K steps from 100 initial conditions, evaluated every 100 emulation steps. Our approach with $L = 3$ consistently sustains a high stability rate during rollout, achieving **93.0**% stability through the $2 \times 10^5$ steps. This greatly surpasses the closest competitor, the 2-Step History method, which only reaches 38.0%, with 2-Step Ahead and our $L = 2$ following. Most other methods fail at the end of rollouts, indicating their limitations in capturing long-term dynamics. These findings highlight the need for our methods that can handle longer temporal dependencies to ensure robust long-term performance.

**Zonal mean and energy spectrum.** We extend our evaluation using (1) the zonal mean and (2) the energy spectrum (see definitions in Appendix A.2). The zonal mean reveals the flow's large-scale organization, namely, jet formation and maintenance. In geophysical turbulence, jets are crucial structures resulting from nonlinear advection, planetary rotation, and dissipation. Capturing the zonal mean is vital for verifying that the emulator preserves these structures over time. As seen in Fig. 4(a), our methods ($L = 2$ and $L = 3$) produce zonal mean profiles within error bands, successfully reproducing the jets, unlike baseline ($L = 1$) and other methods. The energy spectrum illustrates kinetic energy distribution over wavenumbers. In Fig. 4(b), $L = 2$ and $L = 3$ exhibit minimal bias, aligning closely with the reference spectrum, even at small wavenumbers. These findings confirm our model's robustness, maintaining long-term stability while accurately reflecting the system's statistical properties.

## 5.2 Ablation Studies

**Ablation on model hierarchy.** A key aspect of our model is its ability to process and estimate hierarchical representations of the data. To further examine this design, we analyze PCA autocorrelation to capture temporal structure and the energy spectrum to evaluate frequency-domain accuracy. Computation details are provided in the Appendix A.3. The results are shown in Fig. 5 and Fig. 6. Our model with $L = 3$ preserves temporal patterns more effectively and provides more accurate estimates of high-frequency components compared to $L = 2$ and $L = 1$.

**Ablation on downsampling factor.** Our system eases the future prediction tasks by initially assessing a downsampled version of the future frame. The downsampling ratio plays an important role in balancing estimation challenges and available information for next frame prediction. We

| Rollout Step | 1 | 10 | 25 | 50 | 75 | 100 |
|---|---|---|---|---|---|---|
| $L = 1$ ($\ell_1 + \ell_2$ loss) | 5.60e−04 (1.15e−04) | 1.29e−02 (4.87e−03) | 8.04e−02 (3.45e−02) | 3.12e−01 (1.15e−01) | 6.19e−01 (2.08e−01) | 9.38e−01 (2.88e−01) |
| $L = 1$ (Sobolev loss) | 7.01e−04 (4.07e−04) | 1.64e−01 (8.54e−02) | 8.76e−01 (2.45e−01) | 1.40e+00 (2.30e−01) | 1.64e+00 (2.56e−01) | 1.69e+00 (2.51e−01) |
| $L = 3$ ($\ell_1 + \ell_2$ loss) | 5.50e−04 (1.20e−04) | 6.59e−03 (2.50e−03) | 3.37e−02 (1.41e−02) | 1.40e−01 (6.16e−02) | 3.24e−01 (1.18e−01) | 5.92e−01 (1.84e−01) |
| $L = 3$ (Sobolev loss) | **6.49e−04 (3.03e−04)** | **1.45e−02 (6.35e−03)** | 7.28e−02 (3.80e−02) | 2.64e−01 (**1.42e−01**) | 5.74e−01 (2.78e−01) | 8.83e−01 (3.21e−01) |

Table 3: **Comparison of rollout MSEs with varied training objectives**. We train the models using (1) standard $\ell_1 + \ell_2$ loss functions; (2) Sobolev loss from Li et al. [25].

| Method | Seconds per iter. | 1-step MSE | 25-step MSE | 50-step MSE | Zonal mean error |
|---|---|---|---|---|---|
| Baseline: $L = 1$ | 0.2067 | 5.60e−04 (1.15e−04) | 8.04e−02 (3.45e−02) | 3.12e−01 (1.15e−01) | 4.20 (2.36) |
| Pushforward [18] | 0.2834 | 1.08e−03 (2.26e−04) | 9.16e−02 (4.62e−02) | 3.19e−01 (1.34e−01) | 1.07 (0.44) |
| Ours: $L = 3$ | 0.2204 | **5.50e-04 (1.20e-04)** | **3.37e-02(1.41e-02)** | **1.40e-01 (6.16e-02)** | **0.63 (0.58)** |

Table 4: **Comparison of training efficiency and multi-step forecasting errors.** Our $L = 3$ method yields better short-term accuracy and preserves better structure coherence, compared to the $L = 1$ baseline and the pushforward unrolling [18].

| | $512 \times 512$ resolution, w jet | | | $256 \times 256$ resolution w/o jet | | |
|---|---|---|---|---|---|---|
| Rollout Step | 25 | 50 | 75 | 25 | 50 | 75 |
| Baseline: $L = 1$ | 2.16e−01 (8.77e-02) | 6.82e−01 (2.55e−01) | 1.13e+00 (3.54e-01) | 2.59e−01 (1.09e-01) | 8.44e−01 (2.88e−01) | 1.42e+00 (3.92e-01) |
| Ours: $L = 3$ | **5.75e-02 (3.04e-02)** | **1.81e-01 (9.02e-02)** | **3.89e-01 (1.83e-01)** | **1.28e-01 (5.13e-02)** | **5.12e-01 (1.76e-01)** | **9.98e-01 (2.55e-01)** |

Table 5: **Mean squared error on different datasets for short-term rollout.** We report mean (standard deviation) across 100 trials of short-term rollout with different initial conditions.

ablate our two-level system across various downsampling rates of $r^1 \in [2, 4, 8, 32]$. Results presented in Table 2 demonstrate that $r^1 = 8$ provides the best performance for nearly all prediction steps.

**Ablation on training objectives.** In addition to the standard $l1$ and $l2$ losses, we further provide results for experiments with the Sobolev loss, proposed in Li et al. [25]. As shown in Table 3, across different training objectives, our $L = 3$ method consistently yields significant improvements compared to the $L = 1$ baseline.

**Comparison on training efficiency and rollout method.** Unlike multi-step unrolling methods [18, 20], our approach achieves greater computational efficiency while enhancing both short-term accuracy and long-term coherence. As illustrated in Table 4, it incurs only about a 10% higher cost per iteration relative to the $L = 1$ baseline, yet substantially reduces errors across both short- and long-horizon metrics. In contrast, the pushforward method [18] mitigates distributional shift at the expense of higher computational cost and degraded short-term accuracy.

### 5.3 Performance on Diverse Datasets

We further evaluate the robustness of our approach across multiple datasets and spatial resolutions. In addition to the jet-forming regime with $\beta = 20$ that mimics the influence of Earth's rotation, we consider an eddy configuration ($\beta = 0$). As shown in Fig. 6, our approach ($L = 3$) consistently produces realistic and accurate flow evolutions. In contrast, the baseline model ($L = 1$) shows large residuals after a short rollout. This performance gap is further quantified in Table 5, where the MSE of $L = 1$ grows rapidly over time, while $L = 3$ maintains low and stable errors.

## 6 Conclusion

In this work, we present a novel framework for autoregressive modeling of dynamical systems, inspired by implicit time-stepping numerical solvers. Diverging from prior approaches, our method uniquely integrates future-state insights to refine next-step predictions. To enable autoregressive operation, we introduce a temporal hierarchy via spatial compression, coupled with an encoder-decoder architecture. This design organizes multiscale spatial-temporal information, empowering the encoder to distill critical features and the decoder to adaptively address prediction challenges across scales. Evaluated on turbulent flow systems, our framework achieves substantial error reduction while maintaining computational efficiency. While our current focus centers on deterministic modeling, the architecture's flexibility suggests promising extensions to generative paradigms (e.g., diffusion models) for probabilistic forecasting and uncertainty quantification.

**Broader Impacts.** While better emulators for dynamical systems could be used in a wide range of applications, we foresee no direct negative societal impacts.

## Acknowledgements

RJ, PL, and RW gratefully acknowledge the support of AFOSR FA9550-18-1-0166, the Eric and Wendy Schmidt AI in Science Fellowship program, and the Margot and Tom Pritzker Foundation. RJ, PL, RW, and MM gratefully acknowledge the support of the NSF-Simons AI-Institute for the Sky (SkAI) via grants NSF AST-2421845 and Simons Foundation MPS-AI-00010513. KJ and PH were supported by NSF RISE-2425898 and Schmidt Sciences, LLC.

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

# A Experimental details.

## A.1 Further experiments.

| Rollout steps | 1 | 25 | 50 | 75 | 100 |
|---|---|---|---|---|---|
| Baseline : $L = 1$ | 5.60e-04 (1.15e-04) | 8.04e-02 (3.45e-02) | 3.12e-01 (1.15e-01) | 6.19e-01 (2.08e-01) | 9.38e-01 (2.88e-01) |
| FNO [22] (4x Params.) | 1.58e-03 (2.82e-04) | 5.22e-02 (3.12e-02) | 1.94e-01 (9.78e-02) | 4.22e-01 (1.95e-01) | 7.12e-01 (2.24e-01) |
| Spatial Hierarchical | 5.15e-04 (1.14e-04) | 6.59e-02 (3.59e-02) | 2.69e-01 (1.31e-01) | 5.77e-01 (2.39e-01) | 9.28e-01 (3.08e-01) |
| History Hierarchy | 5.11e-04 (1.14e-04) | 6.41e-02 (4.03e-02) | 2.36e-01 (1.23e-01) | 4.99e-01 (2.29e-01) | 7.79e-01 (2.70e-01) |
| 2-step Ahead | 1.06e-03 (2.18e-04) | 4.22e-02 (1.87e-02) | 1.79e-01 (7.70e-02) | 4.11e-01 (1.69e-01) | 7.17e-01 (2.40e-01) |
| 2-step History [19] | 4.84e-04 (1.09e-04) | 5.21e-02 (2.76e-02) | 2.02e-01 (8.99e-02) | 4.42e-01 (1.86e-01) | 7.67e-01 (2.71e-01) |
| 3-step History [19] | **4.53e-04 (9.65e-05)** | 4.28e-02 (2.10e-02) | 1.81e-01 (8.71e-02) | 3.96e-01 (1.57e-01) | 6.81e-01 (2.39e-01) |
| Ours: $L = 2$ | 5.25e-04 (1.15e-04) | 4.02e-02 (1.92e-02) | 1.61e-01 (6.50e-02) | 3.73e-01 (1.51e-01) | 6.55e-01 (2.44e-01) |
| Ours: $L = 3$ | 5.50e-04 (1.20e-04) | **3.37e-02 (1.41e-02)** | **1.40e-01 (6.16e-02)** | **3.24e-01 (1.18e-01)** | **5.92e-01 (1.84e-01)** |

Table 6: **Comparisons to other methods.** We compare to other methods using mean squared error (MSE) for autoregressive roll-out across different lengths. We report the results using average and standard deviation, in parentheses, and demonstrate that ours ($L = 3$) outperforms others in all steps above 1-step MSE. We also show that building models with both spatial and temporal hierarchies enhances the stability of the estimation.

| Rollout steps | 1 | 25 | 50 | 75 | 100 |
|---|---|---|---|---|---|
| Baseline: L=1 | 8.02e-07 (4.89e-07) | 9.25e-06 (4.33e-06) | 1.49e-05 (5.56e-06) | 2.16e-05 (7.40e-06) | 2.93e-05 (1.05e-05) |
| FNO [22] (4x Params.) | **4.90e-07 (1.91e-07)** | 4.93e-06 (1.61e-06) | 9.68e-06 (2.59e-06) | 1.52e-05 (3.71e-06) | 2.14e-05 (4.97e-06) |
| Spatial-Hierarchy | 6.28e-07 (3.96e-07) | 1.03e-05 (4.83e-06) | 1.87e-05 (7.54e-06) | 2.65e-05 (9.94e-06) | 3.48e-05 (1.14e-05) |
| History-Hierarchy | 8.52e-07 (6.56e-07) | 8.95e-06 (5.34e-06) | 1.33e-05 (6.66e-06) | 1.84e-05 (8.33e-06) | 2.42e-05 (9.50e-06) |
| 2-step Ahead | 5.89e-07 (4.10e-07) | **5.88e-06 (3.29e-06)** | 9.94e-06 (4.42e-06) | 1.49e-05 (5.71e-06) | 2.10e-05 (7.75e-06) |
| 2-step History [19] | 9.83e-07 (6.22e-07) | 8.11e-06 (4.29e-06) | 1.13e-05 (4.90e-06) | 1.58e-05 (5.87e-06) | 2.18e-05 (7.06e-06) |
| 3-step History [19] | 5.65e-07 (4.40e-07) | 7.05e-06 (3.93e-06) | 1.09e-05 (5.08e-06) | 1.55e-05 (6.30e-06) | 2.12e-05 (8.01e-06) |
| Ours: L=2 | 7.86e-07 (4.75e-07) | 9.42e-06 (4.91e-06) | 1.24e-05 (5.59e-06) | 1.62e-05 (6.14e-06) | 2.10e-05 (6.24e-06) |
| Ours: L=3 | 6.37e-07 (3.80e-07) | 6.28e-06 (2.90e-06) | **9.29e-06 (3.72e-06)** | **1.28e-05 (4.52e-06)** | **1.76e-05 (5.44e-06)** |

Table 7: **Energy spectrum error comparison.** Our $L = 3$ model achieves significant reduction in energy spectrum error during short-term rollouts (up to 200 steps), outperforming all comparison methods - including a baseline with $4\times$ more parameters - for all prediction horizons beyond single-step forecasting.

**Experimenting with FNO**. All experiments thus far have utilized a UNet architecture with Fourier layers, as described in Section 5. To provide additional comparison, we run $L = 1$ with Fourier Neural Operator (FNO) [55]. Key adaptations include: (1) Using the $\ell_1 + \ell_2$ loss (consistent with our other experiments in Table 6) instead of the default Sobolev-norm objective [25], as the latter caused rapid prediction divergence (even within 200 steps); (2) For $256 \times 256$ resolution data, we choose the Fourier mode number through grid search over $\{64, 96, 128\}$; (3) We use 3 FNO layers, which results in a model with 130M parameters — $4\times$ larger than our $L = 3$ configuration.

**Model Efficiency.** Our model utilizes the UNet's hierarchical framework to efficiently process multi-scale data. As shown in Table 8, compared to the baseline $L = 1$, ours $L = 3$ adds only a few convolutional heads for handling and outputting latent variables $z$, resulting in minimal parameter and inference time overhead.

| | Param Counts (M) | Forward Time (seconds) |
|---|---|---|
| Baseline: $L = 1$ | 34.88 | 0.030 |
| Ours: $L = 3$ | 36.67 | 0.031 |

Table 8: **Model Running Time and Parameters comparison**. On input with $256 \times 256$ resolution, our design ($L = 3$) leverages the inherent hierarchical representation of UNet to process hierarchical latent variables, leading to minimal computational overhead than the baseline ($L = 1$).

## A.2 Evaluation metrics.

**Mean squared error (MSE).** We use

$$\text{MSE}(\boldsymbol{u}_n, \hat{\boldsymbol{u}}_n) = \|\boldsymbol{u}_n - \hat{\boldsymbol{u}}_n\|_2^2$$

as our primary metric to quantify the prediction error for short-term predictions.

**Long-term stability.** To evaluate the long-term stability of predictions, we leverage the system's conserved energy, defined as $E = \frac{1}{2}(v_x^2 + v_y^2)$. As discussed in Section 5.1, we calculate the standard deviation of the ground truth energy values. A trajectory is deemed stable if its energy remains within 5 standard deviations of this reference. This threshold is informed by the observation that all training data lie within 4 standard deviations, while even the true dynamics extrapolated over ten times the training horizon (simulating extreme long-term behavior) do not exceed 5 standard deviations, as shown in Fig. 7. Appendix B.2 provides a detailed visual comparison when the predicted dynamics exceed $\pm 5$ standard deviation range.

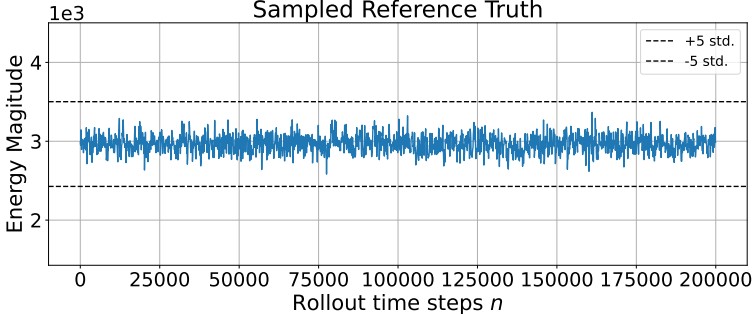

Figure 7: **Energy of true dynamics along extended timescale.** We compute the energy evolution of the true dynamics over 10-times the training dataset length ($2 \times 10^5$ time steps). We show that energy along the sampled true dynamics remains within the $\pm 5$ standard deviation of its mean.

**Fourier spectrum error.** We follow the implementation in py2d to compute the energy spectrum[2]. For short-term evaluation, we compute the mean absolute error of the energy spectrum—the spatial FFT $\mathcal{F}[\boldsymbol{u}_n]$, averaged over $N$ timesteps:

$$\frac{1}{N} \sum_{n=1}^{N} \|\mathcal{F}[\boldsymbol{u}_n] - \mathcal{F}[\hat{\boldsymbol{u}}_n]\|_1.$$

For long-term evaluation (e.g., $N = 2 \times 10^5$), where a ground truth trajectory is unavailable for every new initial condition, and the system loses memory of initial conditions showing an invariant spectrum due to its ergodic properties, we instead compare against a fixed reference spectrum $\mu[\mathcal{F}[\boldsymbol{u}]] := \frac{1}{N} \sum_{n=1}^{N} \mathcal{F}[\boldsymbol{u}_n]$, computed from an extremely long true trajectory. The error is then:

$$\left\| \frac{1}{N} \sum_{n=1}^{N} \mathcal{F}[\boldsymbol{u}_n] - \mu[\mathcal{F}[\boldsymbol{u}]] \right\|_1.$$

**Zonal mean.** The zonal mean of vorticity is computed by averaging vorticity perpendicular to the jets (i.e., along the $x$-direction), shown as:

$$\frac{1}{N N_x} \sum_{n=1}^{N} \sum_{x=1}^{N_x} \boldsymbol{u}_{n,x}.$$

$\boldsymbol{u}_{n,x} \in \mathbb{R}^d$ denotes the vector of the vorticity at $x$-axis. $N_x$ denotes the number of discretization points.

---

[2]https://github.com/envfluids/py2d/blob/main/py2d/spectra.py

### A.3 Experimental setup.

**Model Architecture** We adopt the UNet architecture following the design in [53], with several customizations. For the experiments on $256 \times 256$ resolution data, our encoder consists of 5 groups with latent channel sizes of $[16, 32, 64, 128, 128, 128]$. The spatial resolution is halved after each encoder group, resulting in a $8 \times 8$ resolution at the bottleneck for $256 \times 256$ input images. Each encoder group contains two convolutional blocks, which is made up of two convolutional layers with group normalization and residual connections. The decoder mirrors the encoder and incorporates skip connections from corresponding encoder layers. To improve the model's ability to capture long-range spatial dependencies and multi-frequency signals, we add a Fourier layer to each convolutional block. To balance between accuracy and computational efficiency, we limit the number of frequency modes to 96 for $256 \times 256$ data and apply Fourier convolutions in a depth-wise manner, *i.e.*, without inter-channel communication.

For $512 \times 512$ resolution inputs, we use a similar architecture but add an extra encoder group, resulting in latent channel sizes of $[16, 32, 32, 64, 128, 128, 128]$. In the $L = 3$ setup, we set $r^1 = 8$ and $r^2 = 32$, the same rates used for our $256 \times 256$ resolution experiments. Using these rates, we inject latent codes $z_1^{(1)}$ and $z_2^{(2)}$ into encoder groups with feature resolutions of $64 \times 64$ and $32 \times 32$.

**Data preprocessing.** To stabilize training, we normalize the data using a constant scaling factor such that the resulting values have an approximate standard deviation of one. Specifically, for the dataset with Reynolds number 10000, we apply a dividing factor of 10, while for the dataset with Reynolds number 5000, we use a dividing factor of 6.

**Corner cases.** For models that take multiple temporal frames as input, we simulate the initial rollout setting during training by randomly zeroing out early history frames with a probability of 15%, approximating the absence of pre-initial frames.

**Upsampling projections.** Prior to injection, we upsample the latent variables and apply convolutional layers to match the corresponding encoder channels. Decoding is performed at the same spatial resolutions as the injected latent codes. The same architectural setup as $L = 3$ is used to construct baseline models for Spatial Hierarchy and History Hierarchy. In the $L = 2$ case, we use $r^1 = 32$ and inject $z_1^{(1)}$ into the encoder group at the $64 \times 64$ resolution level.

**Training details.** We process the raw data generated from "py2d" [52] solver. Our emulator predicts vorticity at 0.05 time intervals, representing a $500\times$ coarser temporal resolution than the numerical solver's 0.0001 timestep. We normalize the raw simulation data to approximate a standard normal distribution. Since the data is naturally zero-centered, we simply scale it by constant factors (10 for jet-containing flows and 6 for jet-free flows in Section 5.3) to achieve unit standard deviation.

For all experiments, we use the AdamW optimizer with learning rate at $3 \times 10^{-4}$. We conduct experiments on NVIDIA A100, H100, and L40S GPUs.

# B Additional visualizations.

## B.1 Further visualizations.

We provide additional visualization of short-term rollout in Fig. 8, Fig. 9, Fig. 10, and Fig. 11. The comparison across various datasets and methods show consistent improvements of our methods ($L = 2$ and $L = 3$).

## B.2 Stability.

We present four trials of long-term rollouts across all methods, each with distinct initial conditions. The energy evolution over $2 \times 10^5$ timesteps is shown in Figures 12 and 15, while the corresponding dynamic visualizations appear in Figures 13 through 17.

Our analysis reveals that deviations beyond the $\pm 5$ standard deviations range of the ground truth energy distribution consistently correlate with unstable, exploding dynamics or overly smoothed, averaged patterns. While baseline methods exhibit persistent failures once instability occurs, our $L = 2$ model demonstrates a unique recovery pattern, where the dynamics (and associated energy values) can return to physically plausible states after temporary deviations. This robustness stems

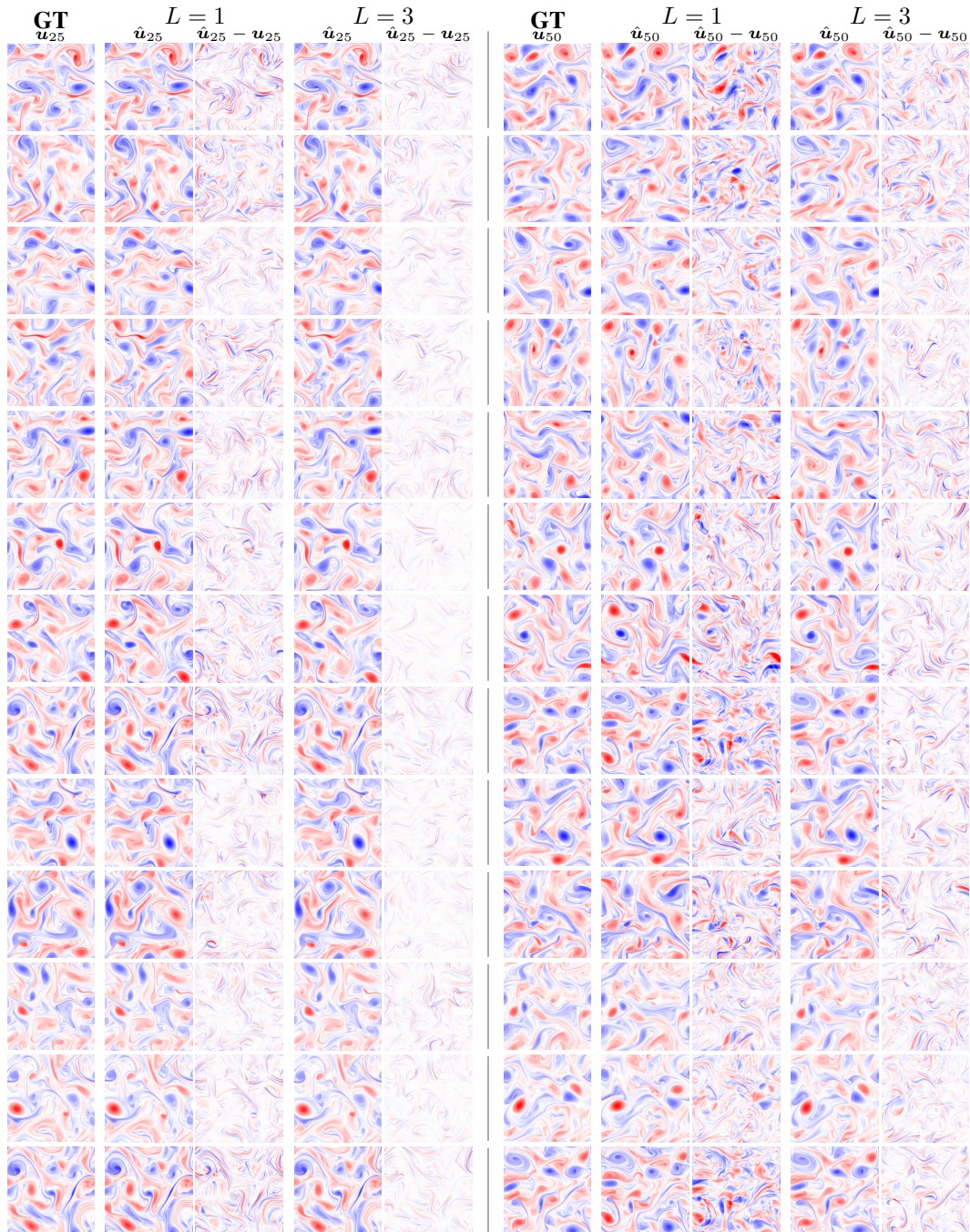

Figure 8: We apply our approach to flow dataset of $Re = 5 \times 10^3$, $256 \times 256$ resolution without zonal jets. Our method ($L = 3$) gives more accurate predictions with lower associated residuals.

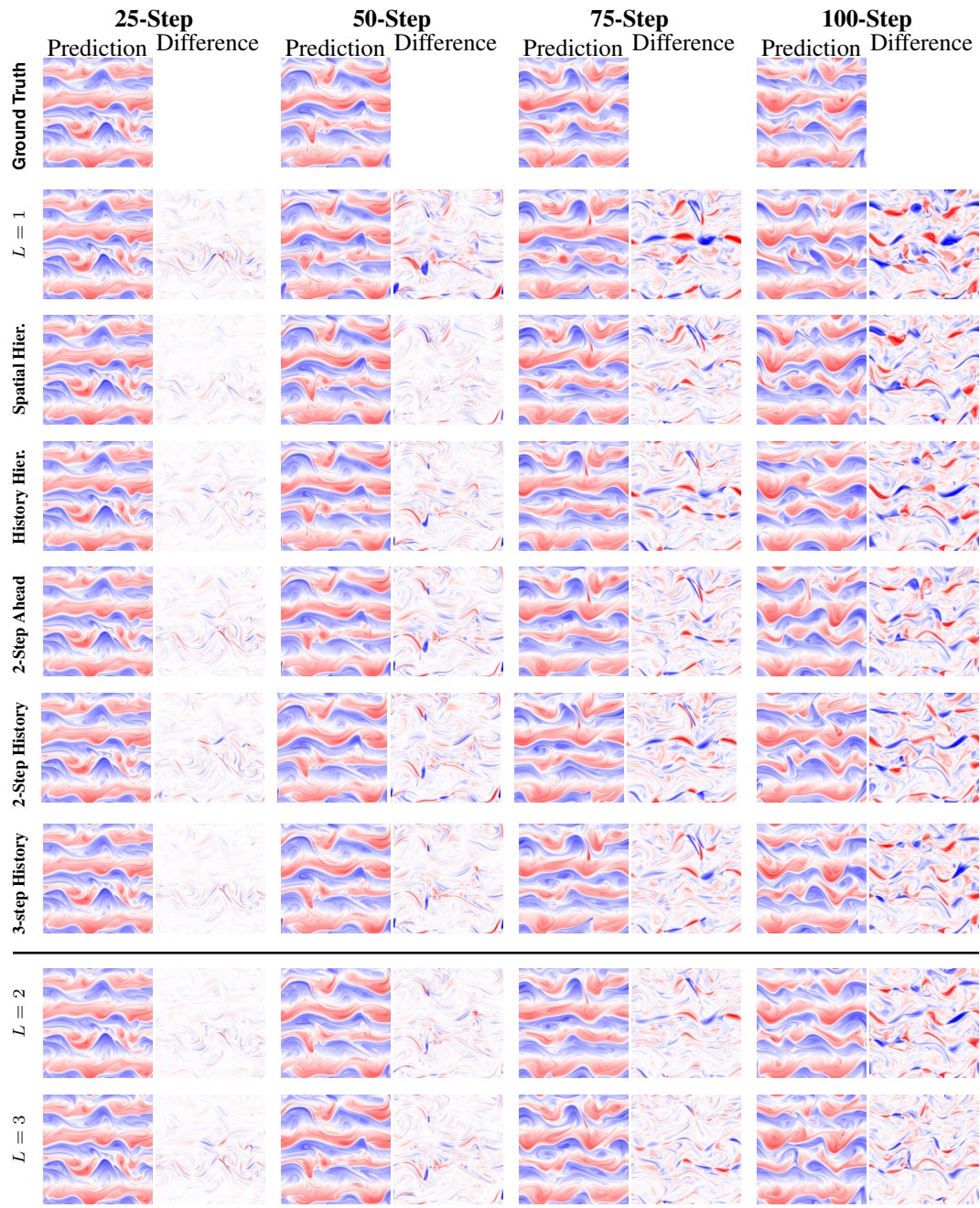

Figure 9: Visualization of prediction and residual to ground truth across methods and prediction steps. Our methods $L = 2, L = 3$ consistently outperform all compared methods.

from the guidance provided by the top-level compressed variables, which help correct errors in fine-grained details.

For stability rate calculations, we conservatively classify a trajectory as unstable if it ever exceeds the predefined $\pm 5$ standard deviations' bounds, regardless of subsequent recovery. This ensures fair comparison across methods.

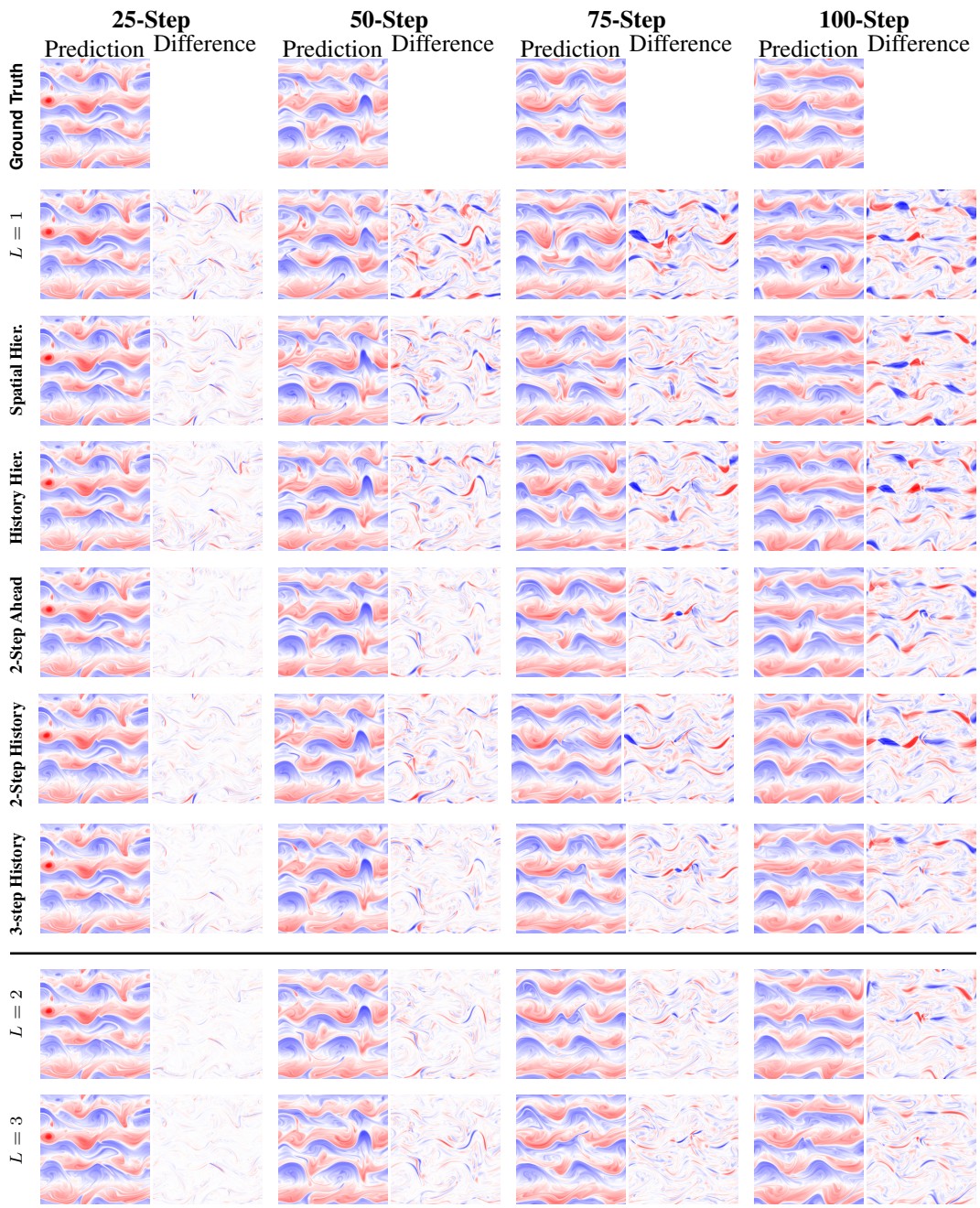

Figure 10: Visualization of prediction and residual to ground truth across methods and prediction steps. Our methods $L = 2, L = 3$ consistently outperform all compared methods.

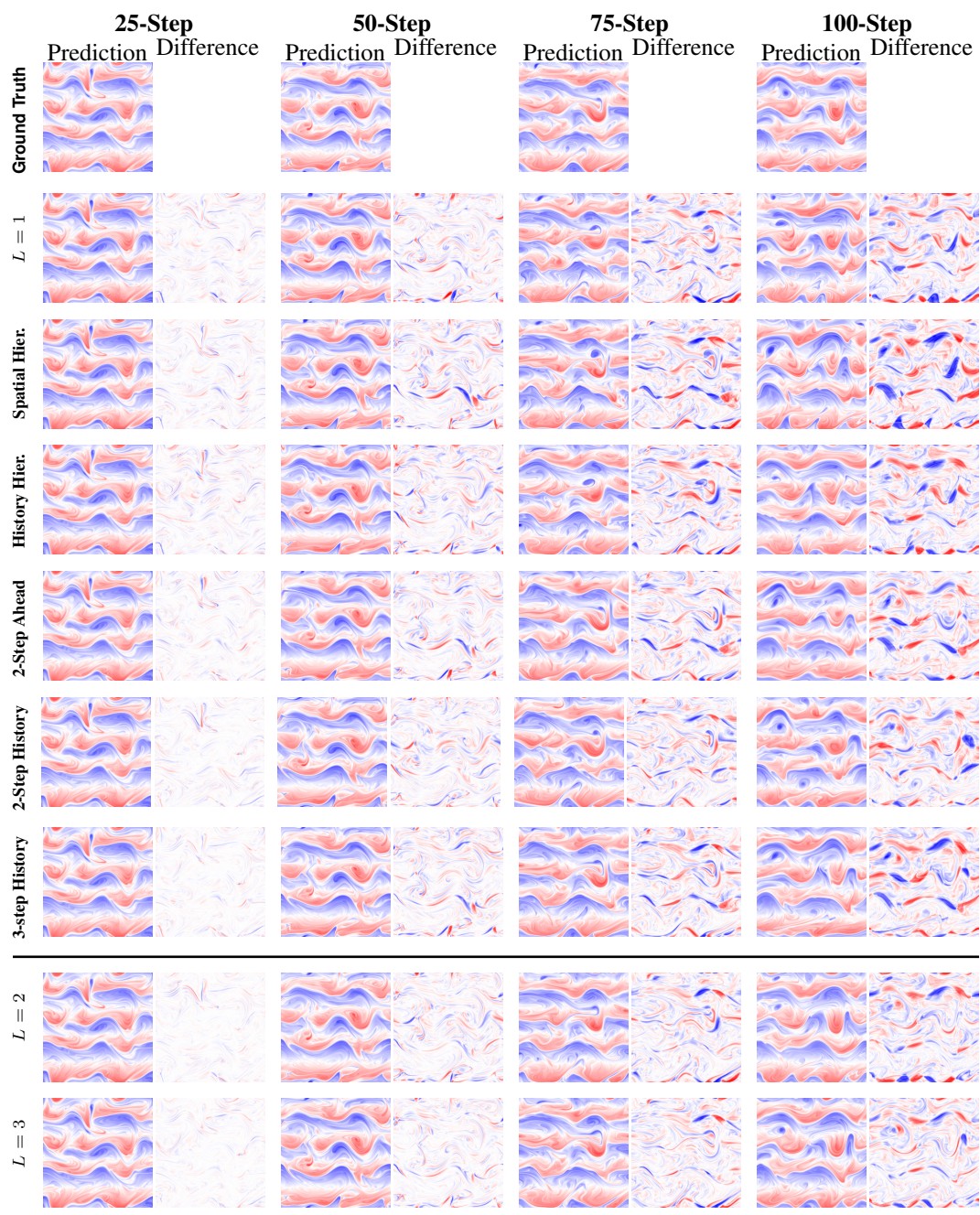

Figure 11: Visualization of prediction and residual to ground truth across methods and prediction steps. Our methods $L = 2, L = 3$ consistently outperform all compared methods.

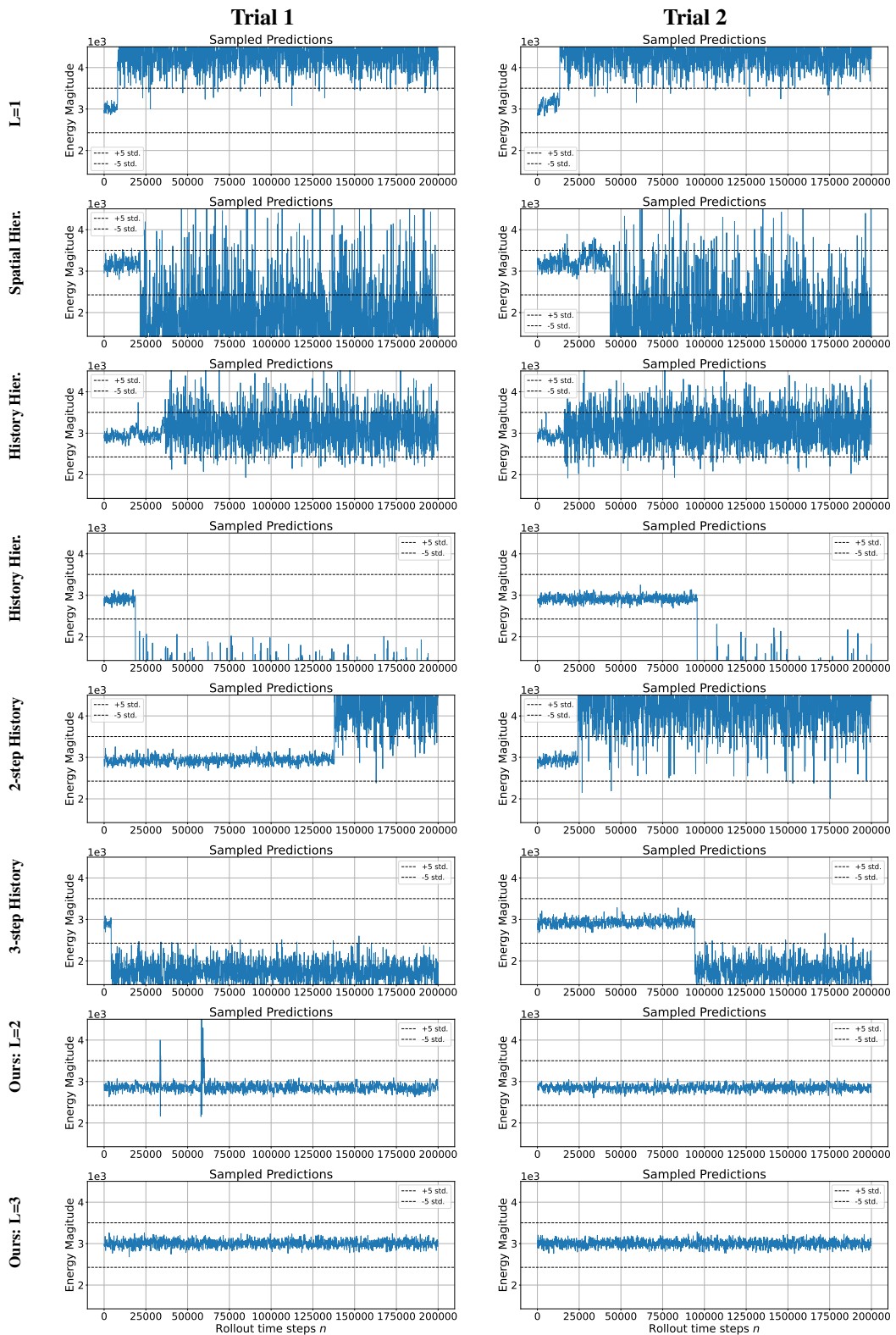

Figure 12: **Energy v.s. rollout time steps of the predicted dynamics.** The dashed horizontal line shows the maximum and minimum values computed using 5 standard deviations from the mean of the true dynamics. Our $L = 3$ method is able to maintain energy along the long-term predictions.

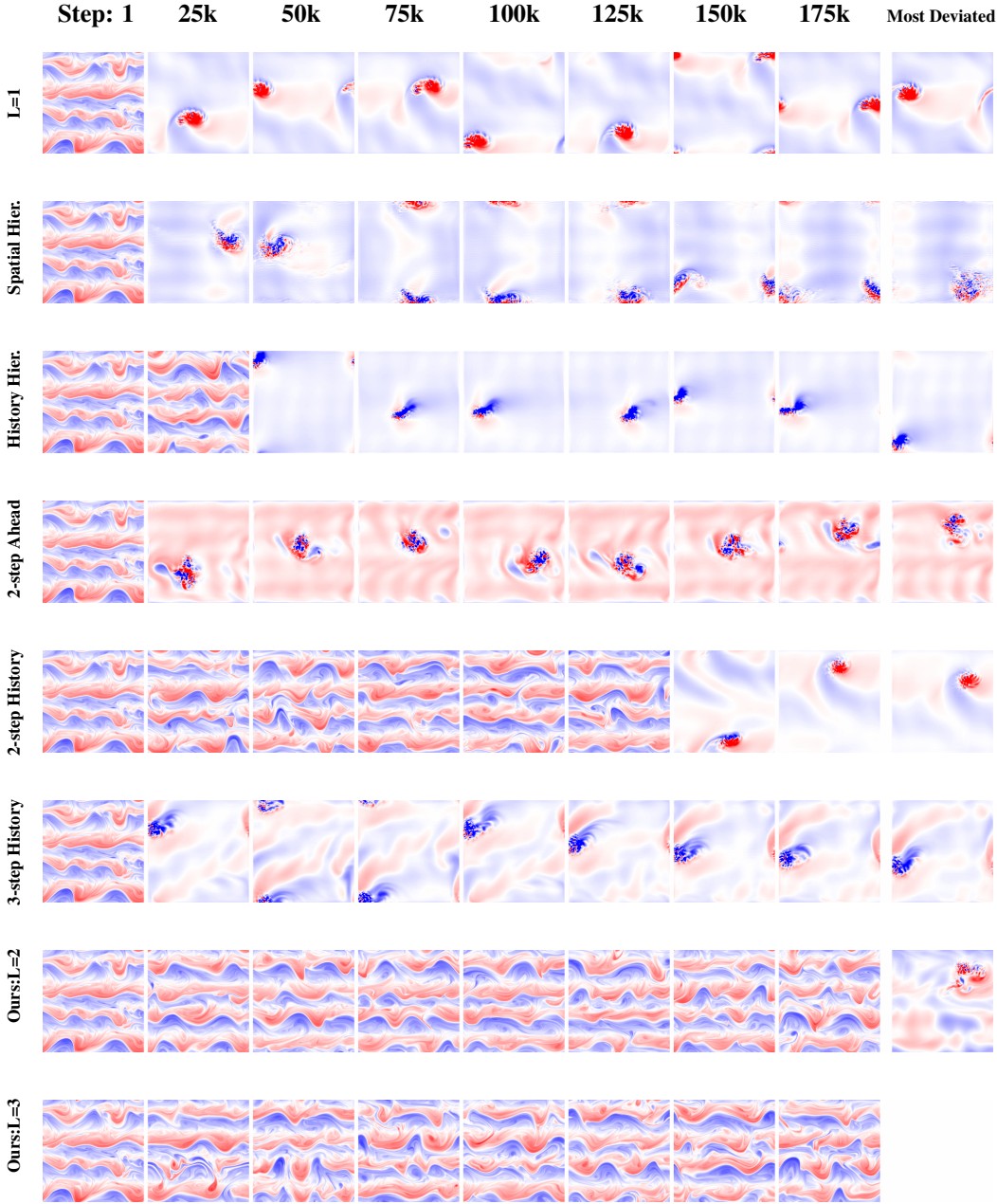

Figure 13: **Long-Term Rollout Visualization.** Long-term dynamics corresponding to Trial 1 in Fig. 12. The rightmost column shows the frame with the maximum deviation—exceeding $\pm 5$ standard deviations from the reference truth statistics, with a blank block indicating that all states along the sampled trajectory remain within this range. When energy predictions fall outside the reference distribution, non-physical dynamics emerge (exploding/averaging artifacts; blank blocks indicate predictions within the $\pm 5$ std. range). Our $L = 3$ model demonstrates superior stability across all comparisons, while $L = 2$ exhibits deviations correlated with energy prediction errors.

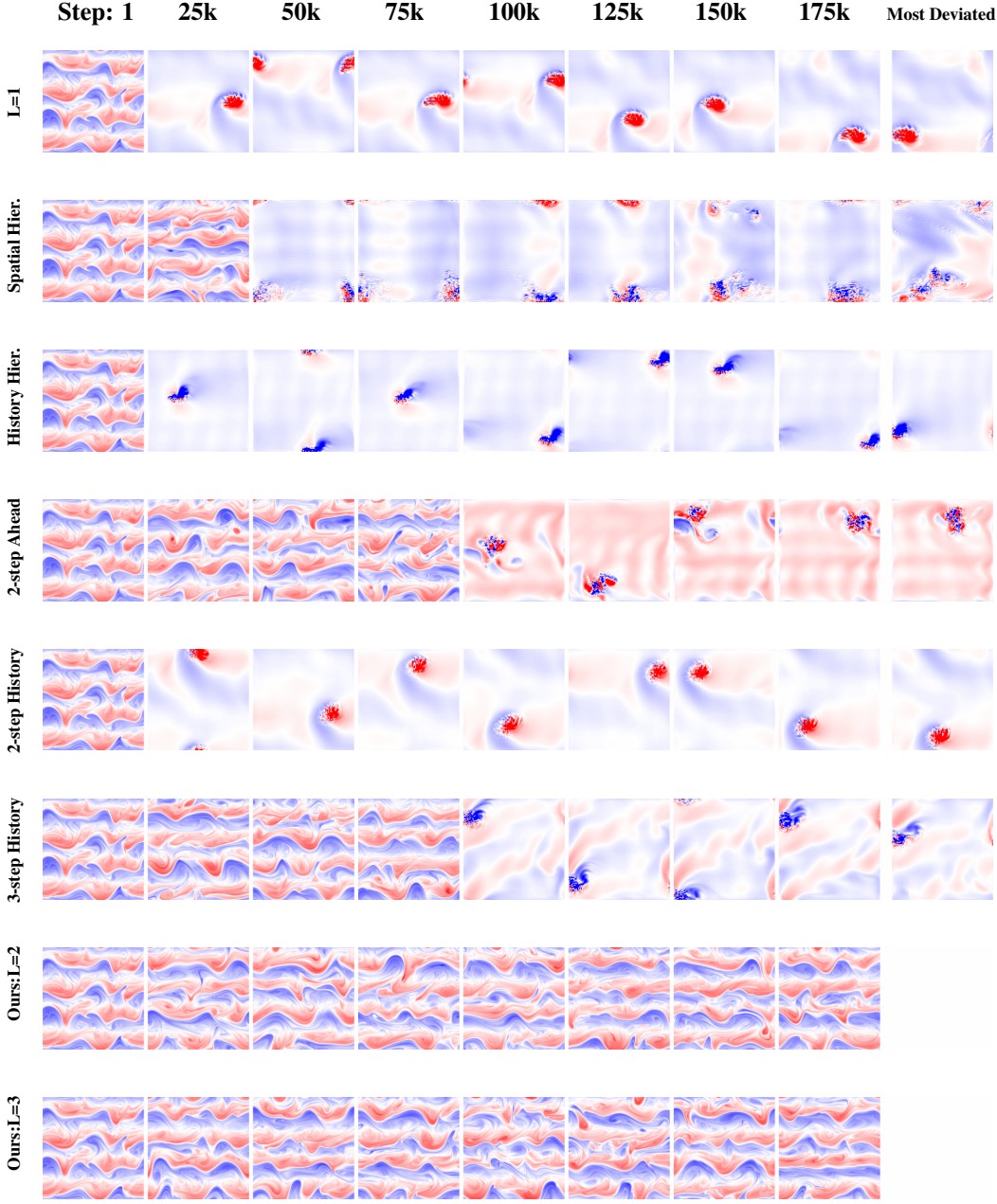

Figure 14: **Visualization of long-term rollout.** We visualize the long-term dynamics corresponds to trial 2 in Fig. 12. The rightmost column shows the frame with the maximum deviation—exceeding $\pm 5$ standard deviations from the reference truth statistics, with a blank block indicating that all states along the sampled trajectory remain within this range. The results show that when the energy is deviated from the reference truth distribution, the dynamics exhibit exploding or averaging non-physical dynamics. Among all comparison methods, our $L = 3$ gives the most stable predictions.

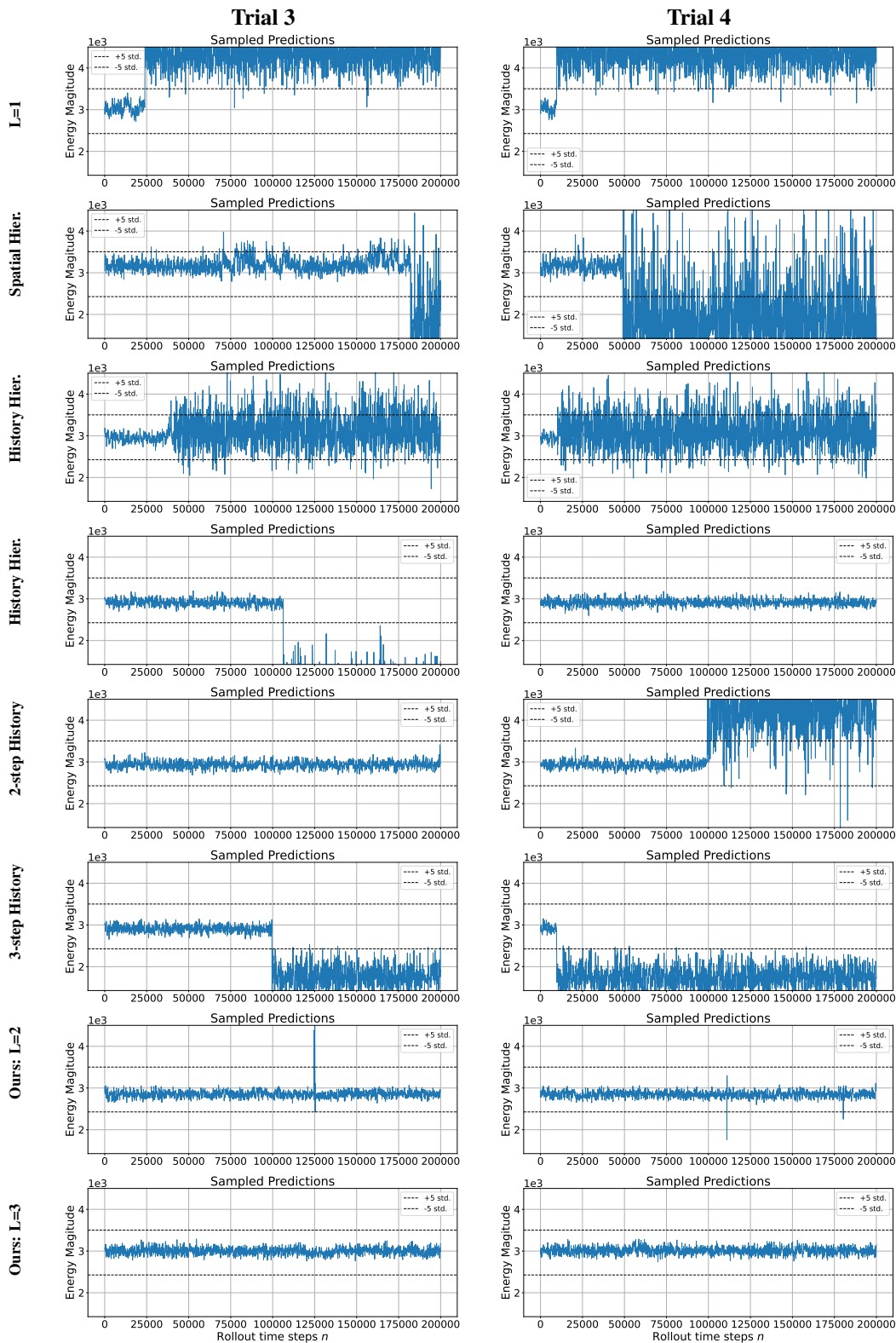

Figure 15: **Energy v.s. rollout time steps of the predicted dynamics.** The dashed horizontal line shows the maximum and minimum values computed using 5 standard deviations from the mean of the true dynamics. Our $L = 3$ method is able to maintain energy along the long-term predictions.

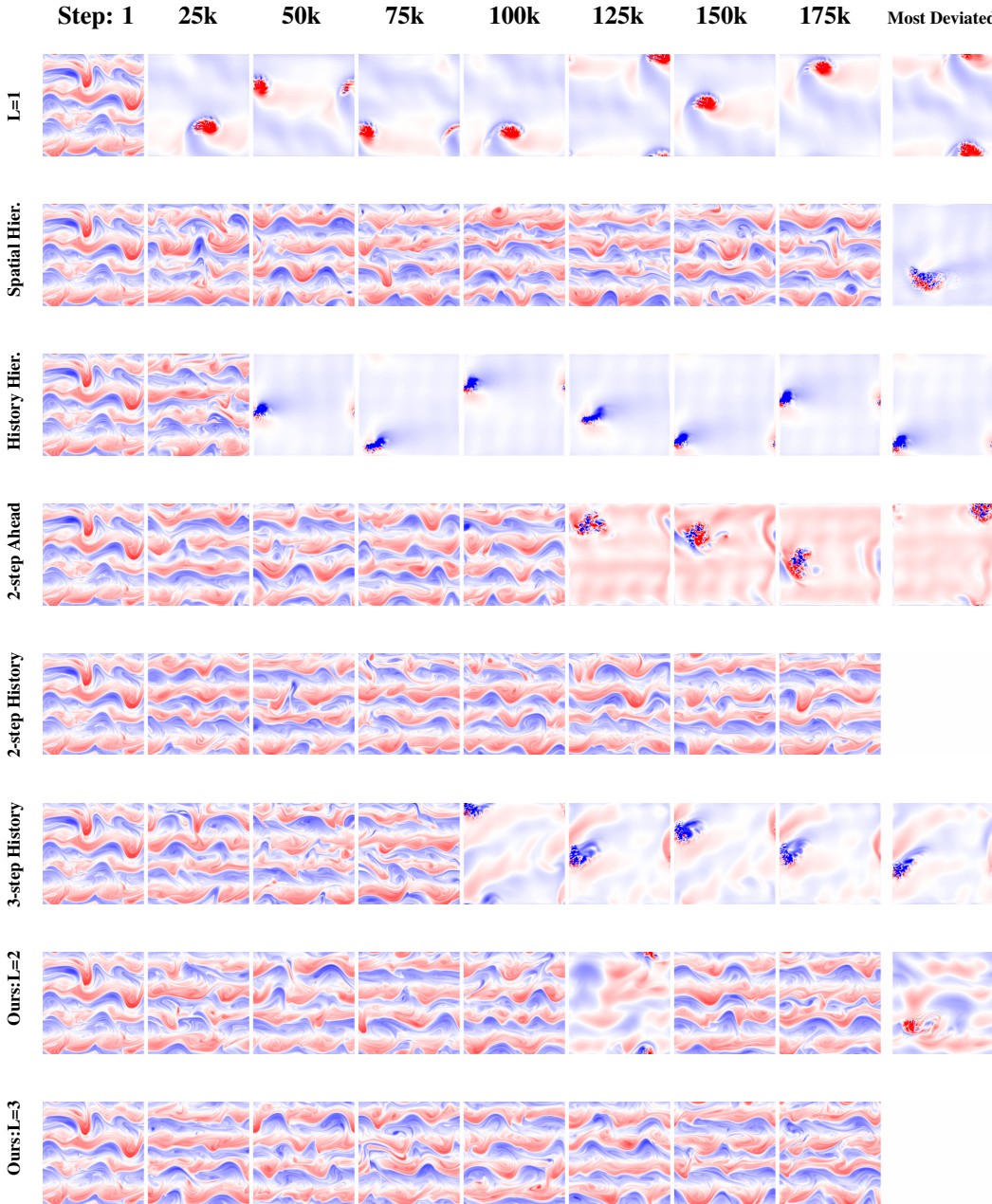

Figure 16: **Visualization of long-term rollout.** We visualize the long-term dynamics corresponds to trial 3 in Fig. 15. The rightmost column shows the frame with the maximum deviation—exceeding $\pm 5$ standard deviations from the reference truth statistics, with a blank block indicating that all states along the sampled trajectory remain within this range. The results show that when the energy is deviated from the reference truth distribution, the dynamics exhibit exploding or averaging non-physical dynamics. Among all comparison methods, our $L = 3$ gives the most stable predictions.

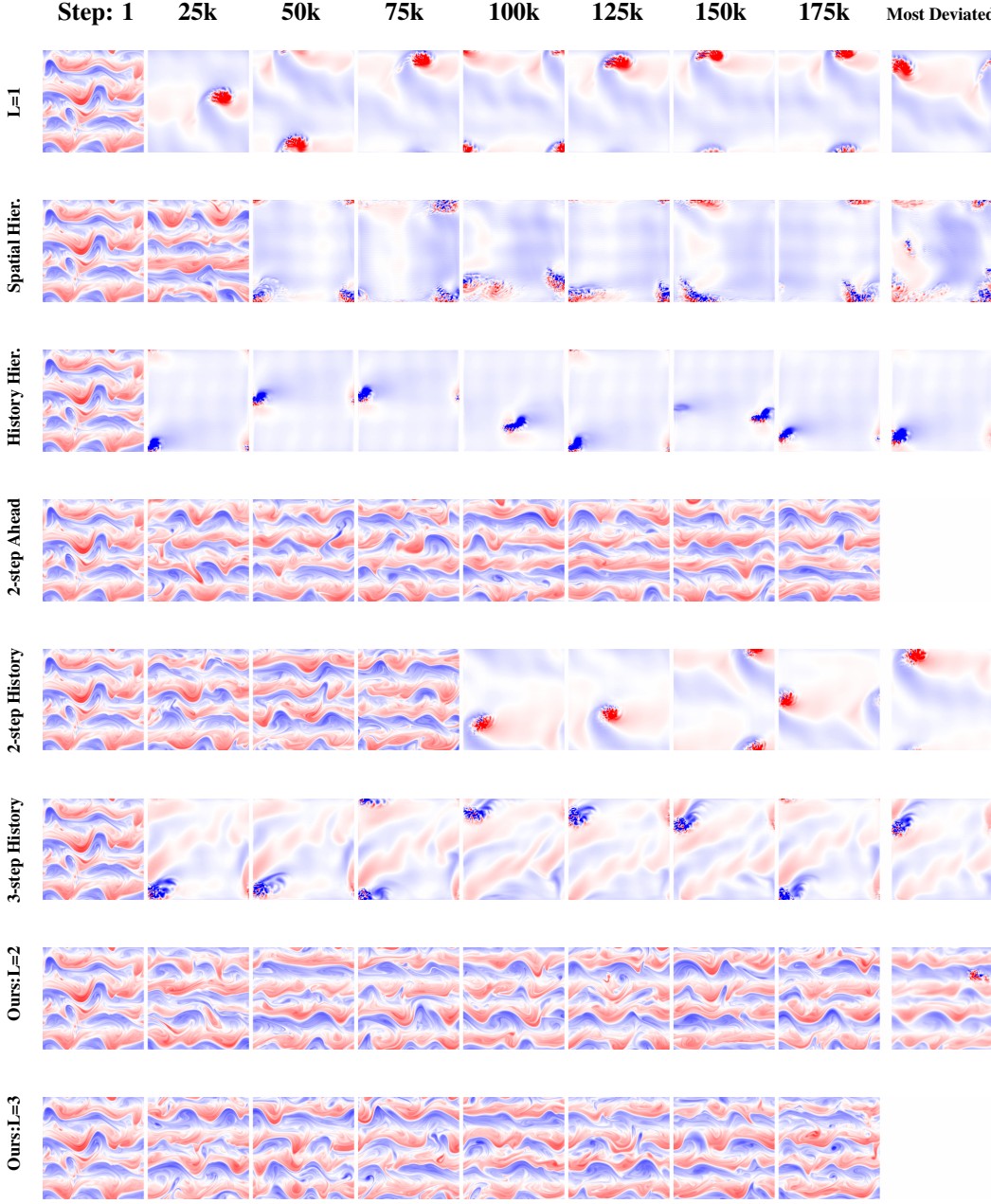

Figure 17: **Visualization of long-term rollout.** We visualize the long-term dynamics corresponds to trial 4 in Fig. 15. The rightmost column shows the frame with the maximum deviation—exceeding $\pm 5$ standard deviations from the reference truth statistics, with a blank block indicating that all states along the sampled trajectory remain within this range. The results show that when the energy is deviated from the reference truth distribution, the dynamics exhibit exploding or averaging non-physical dynamics. Among all comparison methods, our $L = 3$ gives the most stable predictions.

