# OpenReview forum: "Hierarchical Implicit Neural Emulators"
_NeurIPS.cc/2025/Conference — NeurIPS 2025 poster_

### Official Review · Reviewer_vohk · 2025-06-13

**Clarity:** 3
**Significance:** 2
**Originality:** 2
**Rating:** 3
**Confidence:** 3

**Summary:**

The authors present a new method for predicting downsampled future states along with the full-resolution, next state. Additionally, the architecture takes in the full-resolution current state, and a downsampled prediction of a future state. The method performs well on the given benchmarks in short and long term rollouts.

**Questions:**

- It would be interesting to see how the improvement of the implicit scheme varies with the dynamics of the PDE system. For simple systems, explicit schemes are often good enough so would the performance gain be less?

- In Figure 6, the GT for the jet system (top row) for time 25 and 50 look nearly identical (u_25, u_50). Is this the desired behavior?

**Ethical Concerns:**

["NO or VERY MINOR ethics concerns only"]

**Final Justification:**

- Results are good, but limited to a single PDE and architecture
- Extensions to other architectures are not obvious, and the improvements in other PDEs is not clear
- The work has an interesting perspective and with more verification or improvements, it could be a promising method.

**Limitations:**

The authors don’t seem to present any limitations of their work, despite claiming to do so in the conclusion. First, the method seems to be constrained to only Unets, since it doesn’t seem clear how to inject hierarchical information into other architectures. Since the method is constrained to Unets, it doesn’t seem applicable to mesh-based problems. In addition, mesh-based problems don’t admit a consistent downsampling operator for evaluating hierarchical labels. Also, the method would rely on a numerical solver to “kick-start” the surrogate by solving L steps to get the initial, downsampled future states. These aren’t problems with the work, but should be mentioned for clarity.

**Quality:**

2

**Strengths And Weaknesses:**

Strengths

- Predicting multiple future states and taking in representations of a future state is interesting.

- The baselines are done well to test different parts of the proposed method.

Weaknesses

- Perhaps there should be a baseline:  $ u_{n+1}',  u_{n+2}' = f_\theta(u_n, u_{n+1}) $  , to see the effect of the downsampler on the implicit framework. It seems likely that downsampling may be necessary to prevent the network from learning the identity map between $u_{n+1}'$ and $u_{n+1}$.

- There are also a number of stability-enhancing modifications that the authors cite: pushforward/unrolled training, spectral refinement/PDE-refiner. Unrolled training can improve Unet stability by quite a lot [1], and it would be worthwhile to see how implicit methods compare to spectral refinement or unrolled training.

1. Bjoern List, Li-Wei Chen, Kartik Bali, Nils Thuerey, Differentiability in unrolled training of neural physics simulators on transient dynamics, https://www.sciencedirect.com/science/article/pii/S0045782524006960

---

> ### Author Rebuttal · Authors · 2025-07-31
>
> We appreciate your feedback!
>
> ### W1
> > Perhaps there should be a baseline: $u_{n+1}', u_{n+2}' = f_{\theta}(u_{n}, u_{n+1})$ , to see the effect of the downsampler on the implicit framework. It seems likely that downsampling may be necessary to prevent the network from learning the identity map between $u_{n+1}'$ and $u_{n+1}$
>
> Exactly! Without downsampling or any form of information reduction, the model could exploit shortcuts by directly copying high-resolution latent variables. In the proposed setup, where $u_{n+1}', u_{n+2}' = f_{\theta}(u_n, u_{n+1})$, the model could trivially map $u_{n+1}$ to $u_{n+1}'$ via identity, effectively reducing the function to $u_{n+2}' = f_{\theta}(u_n, u_{n+1})$. This is equivalent to our 2-Step History baseline reported in Table 1.
> To verify the empirical effects, we provide results for models without downsampling. We observe that both (1) $L=2$ (No Downsampling) and (2) $L=3$ (No Downsampling) consistently underperform compared to our $L=2$ and $L=3$ models.
>
> | Model                        | 1 Step                | 10th Step            | 25th Step            | 50th Step            | 75th Step            | 100th Step           |
> |------------------------------|-----------------------|----------------------|----------------------|----------------------|----------------------|----------------------|
> | L=1                          | 5.60e-04 (1.15e-04)  | 1.29e-02 (4.87e-03)  | 8.04e-02 (3.45e-02)  | 3.12e-01 (1.15e-01)  | 6.19e-01 (2.08e-01)  | 9.38e-01 (2.88e-01)  |
> | L=2 (No downsample)          | 9.90e-04 (2.14e-04)  | 1.29e-02 (7.02e-03)  | 5.87e-02 (2.89e-02)  | 2.10e-01 (8.89e-02)  | 4.47e-01 (1.76e-01)  | 7.74e-01 (2.52e-01)  |
> | L=3 (No downsample)          | 2.51e-03 (4.41e-04)  | 1.56e-02 (5.11e-03)  | 7.58e-02 (3.15e-02)  | 2.98e-01 (1.31e-01)  | 6.05e-01 (2.30e-01)  | 9.20e-01 (2.94e-01)  |
> | L=2 (ours)                   | **5.25e-04** (1.15e-04)  | 8.11e-03 (4.15e-03)  | 4.02e-02 (1.92e-02)  | 1.61e-01 (6.50e-02)  | 3.73e-01 (1.51e-01)  | 6.55e-01 (2.44e-01)  |
> | L=3 (ours)                   | 5.50e-04 (1.20e-04)  | **6.59e-03** (2.50e-03)  | **3.37e-02** (1.41e-02)  | **1.40e-01** (6.16e-02)  | **3.24e-01** (1.18e-01)  | **5.92e-01** (1.84e-01)  |
>
> Table: Comparison of short-term accuracy on downsampling.
>
> ### W2
>
> > There are also a number of stability-enhancing modifications that the authors cite: pushforward/unrolled training, spectral refinement/PDE-refiner. Unrolled training can improve Unet stability by quite a lot [1], and it would be worthwhile to see how implicit methods compare to spectral refinement or unrolled training.
>
> Thank you for your suggestions. We have run experiments with pushforward training.
> We note that our initial attempt with pushforward training, following the codebase of [r2], did not converge during training—especially in chaotic regimes—as also observed in [r3,r4]. To address this, we introduced a warm-up phase of 50 epochs without pushforward, which significantly improved convergence and overall performance. Consistent with the discussion in [r3], we observe that while pushforward training improves long-term stability, it comes at the cost of short-term accuracy.
>
> | Model                        | 1 Step                | 10th Step            | 25th Step            | 50th Step            | 75th Step            | 100th Step           |
> |------------------------------|-----------------------|----------------------|----------------------|----------------------|----------------------|----------------------|
> | L=2    (ours)        | **5.25e-04** (1.15e-04)  | 8.11e-03 (4.15e-03)  | 4.02e-02 (1.92e-02)  | 1.61e-01 (6.50e-02)  | 3.73e-01 (1.51e-01)  | 6.55e-01 (2.44e-01)  |
> | L=3     (ours)        | 5.50e-04 (1.20e-04)  | **6.59e-03** (2.50e-03)  | **3.37e-02** (1.41e-02)  | **1.40e-01** (6.16e-02)  | **3.24e-01** (1.18e-01)  | **5.92e-01** (1.84e-01)  |
> | Pushforward    | 1.08e-03 (2.26e-04)  | 1.70e-02 (8.63e-03)  | 9.16e-02 (4.62e-02)  | 3.19e-01 (1.34e-01)  | 6.37e-01 (2.71e-01)  | 9.80e-01 (3.36e-01)  |
>
> | Model       | 1k steps               | 10k steps           | 100k steps            |
> |------------------------------|-----------------------|----------------------|----------------------|
> | Pushforward   | 98% | 96% | 88% | 73% |
> | L=3           | **100%** | **100%** | **100%** | **93%** |
>
> Table: Comparison of long-term stability rate with pushforward
>
> Regarding the unrolled training [r1], it introduces higher memory consumption and longer per-epoch training time due to additional forward passes and the requirement of differentiability across the rollouts (e.g., 2-step unrolling requires approximately 2 times larger computational time than our $L=3$ model). We will report these results in the next discussion phase and include the discussions in our refined version.
>
> At last, we want to clarify that both unrolled training [r1] and pushforward training [r2] address the challenge of error accumulation in autoregressive rollouts, while our hierarchical implicit method focuses on exploiting the spatial-temporal structure and iteratively refining latent states using future frames' information.
> These are orthogonal strategies, and combining them (e.g., unrolled training with our method) remains a promising direction, albeit with increased computational cost.
>
> ### Q1
>
> >It would be interesting to see how the improvement of the implicit scheme varies with the dynamics of the PDE system. For simple systems, explicit schemes are often good enough, so would the performance gain be less?
>
> Thank you for the suggestions.
> Our goal is to develop a robust framework for long-term simulation of complex and chaotic systems.. For reference, we have also conducted experiments on a relatively simpler 1D Kuramoto–Sivashinsky PDE.
>
> | L   | 1-step             | 5-step             | 10-step            | 15-step            | 20-step            | 25-step            | 50-step            |
> |-----|--------------------|--------------------|--------------------|--------------------|--------------------|--------------------|--------------------|
> | 1   | 5.43e-06 (1.67e-06) | 2.11e-04 (6.25e-05) | 4.16e-03 (4.30e-03) | 3.88e-02 (4.31e-02) | 1.44e-01 (1.02e-01) | **4.07e-01** (1.76e-01) | **2.03e+00** (1.68e-01) |
> | 3   | **5.21e-06** (1.51e-06) | **1.31e-04** (6.06e-05) | **3.78e-03** (5.09e-03) | **3.28e-02** (4.04e-02) | **1.40e-01** (9.70e-02) | **4.03e-01** (1.88e-01) | **2.00e+00** (1.88e-01) |
>
> Table: Comparison of mean squared error (MSE) on Kuramoto-Sivashinsky
>
> ### Q2
> > In Figure 6, the GT for the jet system (top row) for time 25 and 50 look nearly identical (u_25, u_50).
>
> Thank you for pointing this out! We accidentally uploaded the same figure for $u_{50}$. We will fix this in the final version. We will correct this in the final version. For additional visualizations, please refer to Appendix B, Figures 8–17, which cover various setups beyond Figure 6, including different resolutions, Reynolds numbers, short-term accuracy, and long-term stability.
>
> ### Limitations
> > The authors don’t seem to present any limitations of their work. First, the method seems to be constrained to only Unets, since it doesn’t seem clear how to inject hierarchical information into other architectures... These aren’t problems with the work, but should be mentioned for clarity.
>
> We would first like to clarify a key misunderstanding: our method does not require a numerical solver at inference time to "kick-start" the surrogate. Instead, the model is trained to estimate the full latent hierarchy from a single initial frame. Additional implementation details are provided in Appendix A.3.
>
> Specifically, during the training, with a small probability, we let the model receive partially empty latent variables as input and predict the missing ones in the hierarchy latents.
> For example in $L=2$, with a probability of 15%, we ask the model to take $[u_n, \mathbf{0}]$ as input and output $z_{n+1}^{(1)}$. So that in the evaluation, with one extra forward run, we will have the full latent states $[u_n, \hat{z}_{n+1}^{(1)}]$ to continue the autoregressive rollout.
> All methods are evaluated given only one time frame initial condition. E.g., for generating an $T=1000$ sequence trajectory, our $L=3$ model will forward $1002$ times, where the extra $L-2=3$ forward passes are used to handle these corner cases.
> We will update it in the refined version.
>
> Regarding the architectural scope, we do not consider the reliance on U-Net a limitation. U-Net is a highly flexible and widely adopted architecture across various domains and data modalities, and its hierarchical structure aligns naturally with our framework. That being said, our method is not inherently restricted to U-Nets and can potentially be extended to other architectures with hierarchical capabilities.
>
> For mesh-based applications, while it is true that designing hierarchical representations requires additional considerations (e.g., defining consistent downsampling operators), prior work has proposed effective strategies such as vertex clustering, remeshing, and spectral downsampling. While adapting our framework to these settings would require additional architectural development, we view this as a promising direction with many exciting developments [r5] rather than a fundamental limitation.
>
>
> [r1] Differentiability in unrolled training of neural physics simulators on transient dynamics, Bjoern List, Li-Wei Chen, Kartik Bali, Nils Thuerey
>
> [r2] Message passing neural PDE solvers, Johannes Brandstetter, Daniel Worrall, and Max Welling
>
> [r3] Pdebench: An extensive benchmark for scientific machine learning, Makoto Takamoto, Timothy Praditia, Raphael Leiteritz, Dan MacKinlay, Francesco Alesiani, Dirk Pflüger, and Mathias Niepert.
>
> [r4] A dynamics-informed diffusion model for spatiotemporal forecasting, Salva Rühling Cachay, Bo Zhao, Hailey Joren, and Rose Yu
>
> [r5] Hao et al., High-Fidelity, Artist-Like 3D Mesh Generation at Scale.

---

> > ### Comment · Reviewer_vohk · 2025-08-01
> > **Thank you for the response**
> >
> > Thank you for the experiments regarding the downsampling ablation and pushforward trick, and for clarifying my questions. The provided data is insightful, but I still am a little hesitant due to the nature of the results.
> >
> > - Based on Figure 3 in the paper, it seems that the pushforward trick already outperforms a L=2 implicit method on stability rate? I understand that it takes more compute during training, but it is generally applicable across architectures (GNN, Transformer, FNO, etc.) and has seen successful deployment. For example, Graphcast uses a 12-step unrolling curriculum to maintain stability on high-resolution climate prediction. It just seems odd that you can get the majority of the stability benefits on the proposed NS system from a simple and established method.
> > - The paper's central claim is that implicit methods improve stability, but L=1 vs. L=3 on KS have around the same error after only 20-25 timesteps, despite the authors claiming KS is simpler.
> >     - If you have the MSE results for KS, is it straightforward to calculate the stability rate (either based on energy or some other criteria)? How do the methods compare?
> >     - Adding on to this, I'm not sure if KS is actually simpler or not, since the error seems to span 6 orders of magnitude over a shorter horizon compared to NS. Based on personal experience, it is a pretty challenging system to stably predict due to the 4-th order dispersive term, but this isn't too concerning of a point.
> > - Perhaps this is just differences in opinions, but claiming that needing a uniform grid (in order to use a Unet) is highly flexible and can be used with various modalities is somewhat puzzling. I respect your view and want to refrain from discussions about opinions (rather than data/experiments), so I will just say that a majority of simulations that engineers rely on are mesh-based due to the computational complexity of a uniform discretization.
> > - No work is perfect, nor should it need to be; it would be nice to at least mention limitations in your paper in accordance with the checklist.
> >
> > I think the work is a good step and is promising, but many of the evaluations are limited to a single architecture and PDE. When comparing to more methods and datasets, it seems the results are not as consistent?

---

> ### Author Response · Authors · 2025-08-06
> **quick updates**
>
> Dear Reviewer Vohk,
>
> Thank you for submitting your final acknowledgement. We are actively working on the additional comparison experiments you requested and will share our detailed responses shortly after completion. We value your input and ask that you please remain engaged with our updates. Thank you!

---

> ### Author Response · Authors · 2025-08-08
> **P1: Comparison to unrolled training and pushforward methods**
>
> We thank you for your continued feedback and for recognizing our work as a promising step.
> We here provide additional clarifications along with new experimental results, including comparison to unrolled training [r1], the pushforward trick [r2], and the long-term performance on the Kuramoto–Sivashinsky (KS) equation.
>
> >Based on Figure 3 in the paper, it seems that the pushforward trick already outperforms a L=2 implicit method on stability rate?  It just seems odd that you can get the majority of the stability benefits on the proposed NS system from a simple and established method.
>
> **Reduced short-term accuracy**: First, we would like to emphasize that the pushforward trick, as seen in prior works, tends to decrease the short-term accuracy. As shown in the Table 1 below, the pushforward variant yields higher one-step and short-horizon MSE than even the $L=1$ baseline, indicating a trade-off where pushforward gains stability at the cost of accuracy.
>
> **Not able to maintain the physical properties**: To provide a comprehensive view of the long-term performance of pushforward, we've evaluated its performance additionally using (1) zonal mean error; and (2) energy spectrum error in Table 2. As shown in Table 2, the pushforward model exhibits larger errors in these metrics compared to our $L=3$ approach. This suggests that although pushforward improves stability rate to some extent, it does not effectively maintain the correct physical invariants of the system in the long term.
>
> > I understand that it takes more compute during training, but it is generally applicable across architectures (GNN, Transformer, FNO, etc.) and has seen successful deployment. For example, Graphcast uses a 12-step unrolling curriculum to maintain stability on high-resolution climate prediction.
>
> **Increased training time**: Applying unrolled training is not "simple" or yield moderate computational overhead. Even in our experiments on Navier-Stokes $256\times256$ system, using a modest 2-step unrolling doubled the training time per iteration. We also had to reduce the batch size to fit the unrolling into GPU memory. This 2$\times$ slowdown (or more for longer unroll lengths) represents a serious bottleneck for large-scale or high-resolution systems. In real-world applications, such a significant increase in training cost may be prohibitive.
>
> **Not able to maintain the physical property**: Moreover, in prior works, we've seen that the unrolled training can lead to blurring/smoothing of the predictions (poor spectrum). This has also been verified in our experiments: the unrolled training model produced noticeably poorer energy spectra, while our implicit $L=3$ method maintains a much more accurate energy spectrum.
>
> |Rollout Steps|1|25|50|75|100|
> |:--:|:--:|:--:|:--:|:--:|:--:|
> |Baseline : $L=1$ | 5.60e-04 (1.15e-04) | 8.04e-02 (3.45e-02) |3.12e-01 (1.15e-01) | 6.19e-01 (2.08e-01) | 9.38e-01 (2.88e-01)|
> |Ours: $L=3$ | **5.50e-04** (1.20e-04) | **3.37e-02** (1.41e-02) | **1.40e-01** (6.16e-02) | **3.24e-01** (1.18e-01) | **5.92e-01** (1.84e-01)|
> |Pushforward[r2]| 1.08e-03 (2.26e-04) | 9.16e-02 (4.62e-02) | 3.19e-01 (1.34e-01) | 6.37e-01 (2.71e-01)|9.80e-01(3.36e-01)|
> |Unrolled training[r1]| 5.94e-04 (1.32e-04) | 9.96e-02 (5.04e-02) | 4.06e-01 (1.81e-01) | 7.73e-01 (2.70e-01)|1.08e+00(3.01e-01)|
>
> Table 1: Short-term rollout performance using Mean Squared Error (MSE) on the 2D Navier-Stokes system.
>
> |Metrics |Baseline : $L=1$ |Ours : $L=3$ |Pushforward [r2] |Unrolled training [r1]|
> |:--:|:--:|:--:|:--:|:--:|
> |Zonal Mean Error |4.20 (2.36)|**0.63 (0.58)**|1.07 (0.44) |3.54 (1.9)|
> |Energy Spectrum Error |2.48e+02 (8.93e+00)|**3.91e+00 (1.80e-01)**|2.41e+01(1.06e+00)|3.07e+01(6.03e-01)|
>
> Table 2: Long-term rollout performance on Navier-Stokes, measured by Zonal Mean and Energy Spectrum.
>
> These results demonstrate that our approach not only provides better stability, but also delivers superior accuracy over both short- and long-term rollout horizons, without introducing heavy computational overhead.

---

> ### Author Response · Authors · 2025-08-08
> **P2: Short-term accuracy, long-term stability, and extension to mesh-based data**
>
> > The paper's central claim is that implicit methods improve stability, but L=1 vs. L=3 on KS have around the same error after only 20-25 timesteps, despite the authors claiming KS is simpler. If you have the MSE results for KS, is it straightforward to calculate the stability rate (either based on energy or some other criteria)? How do the methods compare?
>
> **Short-term accuracy & long-term stability**: Our core claim is that the implicit multi-step method (higher $L$) improves short-term accuracy while also enhancing long-term stability. We apologize if our discussion of the KS results was unclear. We provide below a detailed comparison of the Kuramoto–Sivashinsky system to highlight both metrics.
>
> It is true that in a chaotic system like KS, prediction errors will grow with the rollout horizon for all methods, eventually reaching a point where the system’s state becomes effectively unpredictable. In our KS setup, we intentionally use a large spatial domain (length 100) to produce strongly chaotic dynamics. Along with a large time step $\Delta t = 2$, very long-term errors (e.g., beyond 20–25 steps) will inevitably grow, and the difference between the $L=1$ and $L=3$ models diminishes at extremely large rollouts.
>
> The stability rate cannot be extended to the KS system as its energy is not guaranteed to be conserved, in contrast to the Navier-Stokes system.
> To provide more insights on the long-term performance, we add one metric, the energy spectrum error, defined as ($ || \frac{1}{N} \mathcal{F} (E_n) - \mu [\mathcal{F} (E) ||_1$), where $\mathcal{F}$ denotes the spatal FFT and $E_n$ represents the energy at time state $n$, to reflect the capacity to preserve the phsycial property.
>
>
> | L   | 1-step MSE | 5-step MSE  | 10-step MSE  | 15-step MSE  | 20-step MSE  | 25-step MSE | 50-step MSE | [0, 10000]-steps spectrum error |
> |-----|-------------|------------|-----------|---------|-----------|--------------------|--------------------|-------------------|
> | 1   | 2.01e-07 (6.18e-08) | 7.80e-06 (2.31e-06) | 1.54e-04 (1.59e-04) | 1.44e-03 (1.59e-03) | 5.33e-03 (3.77e-03) | **1.51e-02** (6.51e-03) | **7.51e-02** (6.21e-03) |1.32e+01 (1.04e+00) |
> | 3   | **1.93e-07** (5.59e-08) | **4.85e-06** (2.24e-06) | **1.40e-04** (1.88e-04) | **1.21e-03** (1.49e-03) | **5.18e-03** (3.59e-03) | **1.49e-02** (6.96e-03) | **7.40e-02** (6.95e-03) | **1.26e+01 (1.05e+00)** |
>
> Table 3: Comparison on the Kuramoto-Sivashinsky system using (1) mean squared error (MSE) evaluated on uniformly normalized values, and (2) energy spectrum error.
>
> As shown in Table 3, our method achieves up to 37.82% error reduction on 5-step MSE, continues to make improvements over extensive rollouts for up to 25 steps, and has a 3.79% reduction in energy spectrum error over a 10000-step rollout. Note that we recalculate the MSE using uniformly normalized values for easier reference.
>
> > Perhaps this is just differences in opinions, but claiming that needing a uniform grid (in order to use a Unet) is highly flexible and can be used with various modalities is somewhat puzzling. I respect your view and want to refrain from discussions about opinions (rather than data/experiments), so I will just say that a majority of simulations that engineers rely on are mesh-based due to the computational complexity of a uniform discretization.
>
> We agree that mesh-based architectures have their important roles in real-world applications. Even within our UNet, we made adaptations according to [r5] by incorporating the spectral convolution from FNO [r6] to better handle high-frequency signals when we tried to improve the baselines. And this effort shows that our method can work well with one of the key design parts of FNO -- a meshgrid-based architecture, for which we believe we could easily retrain all the models and configurations using such architectures.
>
> More importantly, the core ideas of our approach are (a) the hierarchical representations and (b) the recursive prediction of multiple future time points over the hierarchy. These ideas are both compatible with a mesh-based representation, and indeed, there are many papers on multi-scale or hierarchical representations of mesh-based data. Thus, the core ideas of our method could be explored in other settings, even though the implementation would need to change.

---

> ### Author Response · Authors · 2025-08-08
> **P3: Discussion of limitation**
>
> > No work is perfect, nor should it need to be; it would be nice to at least mention limitations in your paper in accordance with the checklist.
>
> We appreciate your openness. We agree that there are still promising additional directions to pursue; nevertheless, our submission represents an important advance in training stable neural operators.
> In fact, in our introduction, we emphasize that our goal is to increase the short-term accuracy along with improving the long-term stability for complex, nonlinear, and multi-scale systems. And this was the main reason we focused on the Navier-Stokes equation, with up to $10^4$ Reynolds number, far surpassing the difficulty level of the majority of works on the same dataset.
>
> Again, we greatly appreciate your sharing with us your expertise in mesh-based designs. We will add this to the limitation checklist. Thank you!
>
> [r1] Differentiability in unrolled training of neural physics simulators on transient dynamics, Bjoern List, Li-Wei Chen, Kartik Bali, Nils Thuerey,
>
> [r2] Message passing neural pde solvers, Johannes Brandstetter, Daniel Worrall, and Max Welling
>
> [r3] Pdebench: An extensive benchmark for scientific machine learning, Makoto Takamoto, Timothy Praditia, Raphael Leiteritz, Dan MacKinlay, Francesco Alesiani, Dirk Pflüger, and Mathias Niepert.
>
> [r4] A dynamics-informed diffusion model for spatiotemporal forecasting, Salva Rühling Cachay, Bo Zhao, Hailey Joren, and Rose Yu
>
> [r5] U-FNO—An enhanced Fourier neural operator-based deep-learning model for multiphase flow, Gege Wen, Zongyi Li, Kamyar Azizzadenesheli, Anima Anandkumar, Sally M. Benson
>
> [r6] Fourier Neural Operator for Parametric Partial Differential Equations, Zongyi Li, Nikola Kovachki, Kamyar Azizzadenesheli, Burigede Liu, Kaushik Bhattacharya, Andrew Stuart, Anima Anandkumar

---

> ### Author Response · Authors · 2025-08-08
> **Waiting for feedback**
>
> Dear Reviewer Vohk,
>
> We kindly wish to draw your attention to our response above, which addresses the concerns you raised. Thank you.

---

> > ### Comment · Reviewer_vohk · 2025-08-08
> > **Reply to Response**
> >
> > Thank you for the effort in providing your response. I will take this in consideration in the final evaluation.

---

> > > ### Author Response · Authors · 2025-08-08
> > > **thank you**
> > >
> > > Thank you for your acknowledgment. Please let us know if you have remaining concerns. Thanks!

---

> ### Author Response · Authors · 2025-08-08
> **updates to Table 1 & 3**
>
> Dear reviewer vohk,
>
> We would like to leave a brief note that we spotted an error when pasting our results for Tables 1 and 3. We've now corrected them. Thank you!

---

### Official Review · Reviewer_HWeY · 2025-06-25

**Clarity:** 3
**Significance:** 2
**Originality:** 3
**Rating:** 3
**Confidence:** 3

**Summary:**

This paper introduces a Multiscale Implicit Neural Emulator to address the long-term instability and error accumulation problems common in neural PDE solvers. Inspired by the stability of implicit numerical time-stepping methods, the proposed approach leverages a hierarchy of coarse-grained future state predictions to guide next-step forecasting. By conditioning on compressed representations of multiple future time steps, the model enforces long-range temporal coherence and significantly improves stability over extended rollouts. Finally authors demonstrated performance of the proposed method on 2D turbulent fluid simulations, achieving better performance than autoregressive baselines.

**Questions:**

Have you evaluated the performance of your method on other types of PDEs?

**Ethical Concerns:**

["NO or VERY MINOR ethics concerns only"]

**Final Justification:**

While the result is good, the rebuttal does not solve my concerns enough, especially, concern about technical soundness. I have fix my score 3.

**Limitations:**

Yes.

**Paper Formatting Concerns:**

No.

**Quality:**

3

**Strengths And Weaknesses:**

Strengths:

・In my understanding, implicit methods have been a cornerstone of success in numerical analysis. Incorporating such techniques into neural PDE solvers could open new directions for advancement in this field.

・The overall presentation is clear and well-written.

Weaknesses:

・In the experiments, the method is only evaluated on a single PDE — the Navier-Stokes equations. Investigating its scalability and performance on other types of PDEs would strengthen the validation of the approach.

・In numerical analysis, the superiority of implicit methods over explicit ones is supported by various theoretical insights (e.g., order of accuracy, stability criteria). It is not discussed whether such theoretical understanding can be carried over or adapted to the neural setting. Addressing this question would significantly strengthen the technical foundation of the proposed method.

---

> ### Author Rebuttal · Authors · 2025-07-31
>
> We appreciate your feedback!
>
> ### W1 and Q1
> > In the experiments, the method is only evaluated on a single PDE — the Navier-Stokes equations. Investigating its scalability and performance on other types of PDEs would strengthen the validation of the approach.
>
> We would like to emphasize that emulating the Navier-Stokes equations under high Reynolds numbers (e.g., $10^4$), with high-resolution data ($256 \times 256$ and $512 \times 512$), and over very long rollout horizons (up to $2 \times 10^5$ steps) presents an extremely challenging task. Achieving substantial improvements in such a demanding setting is notably difficult and highlights the strength of our approach.
>
> For reference, we have also conducted experiments on a relatively simpler 1D Kuramoto–Sivashinsky PDE. Results are shown below:
> | L   | 1-step             | 5-step             | 10-step            | 15-step            | 20-step            | 25-step            | 50-step            |
> |-----|--------------------|--------------------|--------------------|--------------------|--------------------|--------------------|--------------------|
> | 1   | 5.43e-06 (1.67e-06) | 2.11e-04 (6.25e-05) | 4.16e-03 (4.30e-03) | 3.88e-02 (4.31e-02) | 1.44e-01 (1.02e-01) | **4.07e-01** (1.76e-01) | **2.03e+00** (1.68e-01) |
> | 3   | **5.21e-06** (1.51e-06) | **1.31e-04** (6.06e-05) | **3.78e-03** (5.09e-03) | **3.28e-02** (4.04e-02) | **1.40e-01** (9.70e-02) | **4.03e-01** (1.88e-01) | **2.00e+00** (1.88e-01) |
>
> Table: Comparison of mean squared error (MSE) on Kuramoto-Sivashinsky
>
> ### W2
>
> > In numerical analysis, the superiority of implicit methods over explicit ones is supported by various theoretical insights (e.g., order of accuracy, stability criteria). It is not discussed whether such theoretical understanding can be carried over or adapted to the neural setting. Addressing this question would significantly strengthen the technical foundation of the proposed method.
>
> Our design is inspired by implicit methods and shares a conceptual similarity: to produce the final estimate $\hat{u}_n$, the model performs multiple stages of estimation and refinement through the latent variables $\hat{z}^{(L-1)}_n$, ..., $\hat{z}^{(1)}_n$. This process loosely approximates the behavior of an implicit numerical solver that iteratively refines solutions via root-finding steps. However, since these operations involve transitions across multiple latent spaces, whose structure may not admit a simple closed-form, in contrast to classical implicit solvers, it becomes challenging to derive a rigorous theoretical analysis. We hope that the strong empirical performance demonstrated by our method will motivate future work toward a deeper theoretical understanding of this framework.

---

> > ### Comment · Reviewer_HWeY · 2025-08-02
> > **Reply**
> >
> > Thanks for the thoughtful responses.
> >
> > Regarding W1: Thank you for the additional experiments. Along with the challenges of NV at high Re, they address my concerns.
> >
> > Regarding W2: Thank you for your reply. However, I still have concerns about the technical soundness of the proposed method, which prevents me from giving it a higher score. Applying an implicit approach to a neural solver does not strike me as particularly novel. To reach the level required for acceptance, I think the work needs to be further strengthened from a theoretical perspective, particularly in terms of methodological soundness.
> >
> > Taking these points into account, I will maintain my initial scores.

---

> > > ### Author Response · Authors · 2025-08-08
> > > **Conflations of "technical soundness" with "theoretical motivation"**
> > >
> > > ### W1
> > > > Thank you for the additional experiments. Along with the challenges of NV at high Re, they address my concerns.
> > >
> > > Thank you for your acknowledgement! We are glad that our additional experiments address your concerns.
> > >
> > > ### W2
> > > > Thank you for your reply. However, I still have concerns about the technical soundness of the proposed method, which prevents me from giving it a higher score. Applying an implicit approach to a neural solver does not strike me as particularly novel. To reach the level required for acceptance, I think the work needs to be further strengthened from a theoretical perspective, particularly in terms of methodological soundness.
> > >
> > > We believe there is a fundamental error in conflating "technical soundness" with "theoretical motivation": if something is not "technically sound", that would imply that there is a fundamental flaw in the system architecture or experimental setup.  Much work in deep learning is technically sound, while lacking (or having incomplete) theoretical motivation.
> > >
> > > While we appreciate your emphasis on theoretical analysis, we respectfully disagree that the absence of deeper theoretical development undermines the soundness or contribution of our work, or is deemed necessary regarding the acceptance. Specifically:
> > >
> > > **Novelty beyond traditional numerical solver**: even compared to the classical framework of implicit numerical solvers, our method presents a fundamentally more flexible and expressive direction. Specifically, it introduces (1) a generic formulation that allows latent variables to stay in heterogeneous latent spaces across time; (2) a removal of the need for recursive refinement at each step, enabled by augmented states in autoregressive rollouts; and (3) the use of future-frame information that goes beyond merely guessing the next step. These are nontrivial extensions beyond prior work and enable significantly improved modeling capacity and rollout stability.
> > >
> > > **Strong empirical performance**: we demonstrate the robustness, generality, and effectiveness of our approach through extensive empirical validation across diverse datasets, a wide range of spatial resolutions, and varying Reynolds numbers. Our method consistently outperforms existing baselines, showing significant improvements across all settings. These strong and consistent results underscore the technical soundness and practical impact of our methodology.
> > >
> > > While further theoretical study is certainly valuable, we firmly believe that such analysis is not a prerequisite for validating the contribution or soundness of our work -- especially when supported by strong empirical results and clear methodological advances. In line with the NeurIPS guidelines, we contend that our work offers significant value through its novel design and empirical rigor.
> > >
> > > We hope this clarification addresses your concerns. Thank you!

---

> > > ### Author Response · Authors · 2025-08-08
> > >
> > > Dear Reviewer HWeY,
> > >
> > > We kindly wish to draw your attention to our response regarding your concern about the technical soundness of our work. We hope our clarification adequately resolves your concern. Thank you.

---

### Official Review · Reviewer_PLCM · 2025-06-29

**Clarity:** 2
**Significance:** 2
**Originality:** 2
**Rating:** 4
**Confidence:** 3

**Summary:**

This paper introduces a neural PDE solver, with a claim of the implcit manner, leading to a method for a sliding window prediction. The results presented show the effectiveness of the proposed method. The hierarchical design is not news but play a great role in the method. The analysis of the method with the energy spectrum is insightful and interesting (at least for me).

**Questions:**

1. I wonder where the GT is from. Maybe the analytic solution should be listed for the canonical flow to provide a good view for the readers.
2. Some repeated experiments for the results to provide the bar of the curves.
3. how the latents $z$ should be, I think some more theoretical insights for more complex standard PDEs should be provided.
4. how the disantce metirc $d$ infleuence the problem, what if it is not the l1/l2 norm

**Ethical Concerns:**

["NO or VERY MINOR ethics concerns only"]

**Final Justification:**

Most of my problems about the empirical setting are resolved, including the metric, the cases for the GT reconstruction and training time. But I still concern abut the method from the theoretical prespective. Therefore I just raise my score to 4.

**Quality:**

2

**Strengths And Weaknesses:**

Pros:
1. the analysis for the NS results are great, maybe it is more great if it will be visualized in a sphere.
2. the method shows some insight in the implcit manner in the Neural PDEs.

Cons:
1. the latent state for $z$ can not be theoretically guaranteed. Therefore for some PDEs, there might be not latent state existing to work as the explicit prediction manner.
2. I am not sure for the training time. I guess from the loss function design it should take long.
3. the resolution for the results are not large to fully show the effectiveness of the proposed method.

---

> ### Author Rebuttal · Authors · 2025-07-31
>
> We appreciate your feedback!
>
> ### W1
>
> > the latent state for $z$ can not be theoretically guaranteed. Therefore for some PDEs, there might be not latent state existing to work as the explicit prediction manner.
>
> The notion of the latent space $z$ in our framework is intentionally designed to be flexible, as long as it provides a compressed representation of the original data. While more sophisticated latent spaces—such as semantic-aware representations for video data—can be beneficial for specific domains, we adopt a deliberately simple design where $z$ corresponds to spatially downsampled inputs. In the context of PDEs, this form of representation is quite general and has been widely used, for instance, in multigrid methods. We demonstrate that even with this generic choice of latent space, our model achieves significant improvements over strong baselines.
>
>
> ### W2
>
> >I am not sure for the training time. I guess from the loss function design it should take long.
>
> As shown in Table 6 (Appendix A.1), the configuration with $L=3$ exhibits nearly identical runtime and parameter count compared to the baseline with $L=1$. This efficiency arises from our model’s ability to handle multi-resolution data effectively. Specifically, the multi-level architecture reuses most of the U-Net components from the single-level baseline, and we exploit the inherent multi-resolution structure of the U-Net to process inputs at their corresponding feature resolutions. Please refer to Section 5 for additional implementation details.
>
> Benefited from our design, the model only requires a single forward pass per training iteration. With H100 GPU and training mode with batch size 32, single iteration for $L=1$ takes 0.2067 (0.0017) second and $L=3$ takes 0.2204 (0.0021) second. This demonstrates that our method introduces minimal overhead while enabling more expressive multi-level representations.
>
> | Model                        |Seconds per training step| 1 Step                | 10th Step            | 25th Step            | 50th Step            | 75th Step            | 100th Step           |
> |------------------------------|------------------------------|-----------------------|----------------------|----------------------|----------------------|----------------------|----------------------|
> | L=1                 | **0.2067** (0.0017) | 9.17e-02 (7.96e-03)  | 3.87e-01 (6.17e-02)  | 8.21e-01 (1.16e-01)  | 1.17e+00 (8.22e-02)  | 1.29e+00 (6.15e-02)  | 1.34e+00 (5.62e-02)  |
> | L=2  (ours)                        |0.2134 (0.0017) | **5.25e-04** (1.15e-04)  | 8.11e-03 (4.15e-03)  | 4.02e-02 (1.92e-02)  | 1.61e-01 (6.50e-02)  | 3.73e-01 (1.51e-01)  | 6.55e-01 (2.44e-01)  |
> | L=3  (ours)                        |0.2204 (0.0021)| 5.50e-04 (1.20e-04)  | 6.59e-03 (2.50e-03)  | **3.37e-02** (1.41e-02)  | **1.40e-01** (6.16e-02)  | **3.24e-01** (1.18e-01)  | **5.92e-01** (1.84e-01)  |
>
> Table 1: Comparison of training time and short-term accuracy.
>
> ### W3
> > the resolution for the results are not large to fully show the effectiveness of the proposed method.
>
> Our experiments are conducted on a highly chaotic turbulent flow dataset with a Reynolds number of $10^4$, where we evaluate long-term stability over an extended horizon of more than $2 \times 10^5$ rollout steps. Setting up experiments at a resolution of $256 \times 256$ is already extremely challenging due to the complexity of the dynamics. To further demonstrate the robustness and effectiveness of our approach, we also conduct experiments on higher-resolution data at $512 \times 512$, showing that our proposed method scales well and maintains strong performance in this more demanding setting.
>
> ### Q1
> > I wonder where the GT is from. Maybe the analytic solution should be listed for the canonical flow to provide a good view for the readers.
>
> We use an off-the-shelf numerical solver to generate the ground truth data; please refer to Section 5 for implementation details. Our experiments are conducted on highly challenging cases, many of which, such as those governed by the Navier–Stokes equations, do not have known closed-form solutions.
>
> ### Q2
> > Some repeated experiments for the results to provide the bar of the curves.
>
>
> We report standard deviations for most of our results to capture the variability across predicted trajectories—for example, in Figure 3(a), Figure 4(a, c), and Table 2 in the appendix.  We acknowledge the importance of statistical robustness and will include results from additional repeated runs in the final version to further strengthen our findings.
>
> ### Q3
> > how the latents $z$ should be, I think some more theoretical insights for more complex standard PDEs should be provided.
>
> An effective latent space $z$ should provide a compressed yet informative representation of the data, making it easier to estimate than directly predicting the original data, while still retaining sufficient information for accurate prediction. This principle is closely related to several theoretical frameworks, such as the information bottleneck and predictive coding. We will update the final version of the paper to include a discussion of these connections.
>
> ### Q4
> > how the disantce metirc $d$ infleuence the problem, what if it is not the l1/l2 norm
>
> In addition to the standard $l1$ and $l2$ losses, we here provide results for experiments with the Sobolev loss, proposed in [r1].
> We show across different training objectives, our $L=3$ method consistently yields significant improvements compared to the $L=1$ baseline.
>
> | Model                        | 1 Step                | 10th Step            | 25th Step            | 50th Step            | 75th Step            | 100th Step           |
> |------------------------------|-----------------------|----------------------|----------------------|----------------------|----------------------|----------------------|
> | L=1 ($l1+l2$ loss)                         | 5.60e-04 (1.15e-04)  | 1.29e-02 (4.87e-03)  | 8.04e-02 (3.45e-02)  | 3.12e-01 (1.15e-01)  | 6.19e-01 (2.08e-01)  | 9.38e-01 (2.88e-01)  |
> | L=1 (Sobolev loss)           | 7.01e-04 (4.07e-04)  | 1.64e-01 (8.54e-02)  | 8.76e-01 (2.45e-01)  | 1.40e+00 (2.30e-01)  | 1.64e+00 (2.56e-01)  | 1.69e+00 (2.51e-01)  |
> | L=3  ($l1+l2$ loss)                       | **5.50e-04 (1.20e-04)** | **6.59e-03 (2.50e-03)** |**3.37e-02 (1.41e-02)**  | **1.40e-01 (6.16e-02)**  | **3.24e-01 (1.18e-01)**  | **5.92e-01 (1.84e-01)**  |
> | L=3 (Sobolev loss)           | 6.49e-04 (3.03e-04) | 1.45e-02 (6.35e-03) | 7.28e-02 (3.80e-02) | 2.64e-01 (1.42e-01) | 5.74e-01 (2.78e-01) | 8.83e-01 (3.21e-01)|
>
> Table 2: Comparison of short-term accuracy when trained with different metrics
>
> | Model                        | 1k steps               | 10k steps           | 100k steps            |
> |------------------------------|-----------------------|----------------------|----------------------|
> | L=1 (Sobolev loss)           | 3% | 0% | 0% |
> | L=3 (Sobolev loss)           | **100%** | **99%** | **97%** |
>
> Table 3: Comparison of long-term stability when trained using the Sobolev norm
>
> [r1] Learning Dissipative Dynamics in Chaotic Systems, Zongyi Li, Miguel Liu-Schiaffini, Nikola Kovachki, Burigede Liu, Kamyar Azizzadenesheli, Kaushik Bhattacharya, Andrew Stuart, Anima Anandkumar

---

> ### Comment · Reviewer_PLCM · 2025-08-04
> **Reply for the rebuttal**
>
> Thanks for your reply!
>
> Here are some of my further concerns:
> W1: As far as I understand, the latent z should be the coarse error of the grid explicitly or the similar combination. If so, I think it can be theoretically guaranteed.
> W3: I am not sure whether it is 10^4 is a huge Reynolds and under some cases, it will not provide much turblence. The resolution with some units should be provided and the CFL conditions also should to give more insights about the complexity with high resolution.
>
> I will keep my score considering about the points.

---

> > ### Author Response · Authors · 2025-08-08
> >
> > Dear reviewer PLCM,
> >
> > We would like to kindly draw your attention to our latest response addressing your concern regarding the dataset. Thank you.

---

> > > ### Comment · Reviewer_PLCM · 2025-08-08
> > > **Reply for the rebuttal**
> > >
> > > Thanks for the authors. First, I need to recongize that it's my fault about the turblence. That should be about 4000-5000 for the transition flow. I've no problem and thanks a lot for the clarification. But for the first concern, I am not requesting the authors to proivde the total proof for the NS complex system. At least, for the title/contributions of the paper are not focusing on a certain system, I believe a simple PDE with a classic numerical method guarantee can be tried. We all know it is difficult for the theoreical guarantee for the learning method. But at least, more insights/characteristics should be provided for the practitioner to choose with more confidence under their cases.
> > >
> > > But I promise to raise my score to 4 since I make a mistake and again thanks for your pointing out. At least, one of my conerns is resolved.

---

> > > > ### Author Response · Authors · 2025-08-08
> > > > **thank you**
> > > >
> > > > We thank you for your acknowledgement! We are glad that your concern regarding the dataset has been addressed.
> > > >
> > > > And yes! We hope that our work could help inspire new directions for theoretical developments in the field. Thank you!

---

> ### Author Response · Authors · 2025-08-08
> **P1: Turbulence in Navier-Stokes**
>
> > As far as I understand, the latent z should be the coarse error of the grid explicitly or the similar combination. If so, I think it can be theoretically guaranteed.
>
> Our approach's foundation (both implicit integration and hierarchical coarse graining) are backed by decades of work on numerical analysis and multi-scale modeling. However, adding the theoretical analysis for our work, in the regime of deep learning for nonlinear PDEs are not trivial, especially when the mathematical machinery for such analysis is not mature enough, as seen in recent scientific machine learning (SciML) works.
>
> Moreover, our model’s design advances the classical implicit integration schema. Even compared to the classical framework of implicit numerical solvers, our method presents a fundamentally more flexible and expressive direction. Specifically, it introduces (1) a generic formulation that allows latent variables to stay in heterogeneous latent spaces across time; (2) a removal of the need for recursive refinement at each step, enabled by augmented states in autoregressive rollouts; and (3) the use of future-frame information that goes beyond merely guessing the next step. These are nontrivial extensions beyond prior work and enable significantly improved modeling capacity and rollout stability.
>
> While further theoretical investigation would be valuable, we maintain that such analysis is not a prerequisite for validating our work’s contributions, especially given its strong empirical results and methodological advances. Aligning with NeurIPS guidelines, we emphasize that our work delivers significant value through its novel design and rigorous experiments.
>
> > I am not sure whether it is 10^4 is a huge Reynolds and under some cases, it will not provide much turblence. The resolution with some units should be provided and the CFL conditions also should to give more insights about the complexity with high resolution.
>
> First, note that the governing equations are nondimensionalized, and our grid resolutions ($256\times256$ and $512\times512$ grids) do not carry physical units.
>
> In our ground-truth simulations, the CFL number is approximately 0.05, which is well below the stability threshold of 1.0.
> This ensures the numerical stability of the solver, even under high Reynolds number conditions. Specifically, the numerical solver saves data with a time step of $\Delta t_{\rm numerical} = 0.0001$ to guarantee stability, while our training data for emulators is sampled at a coarser time interval with $\Delta t_{\rm emulator} = 500 \Delta t_{\rm numerical} = 0.05$.
>
> We emphasize that a Reynolds number of 10,000 in our setup **does lead to a flow with clear turbulent characteristics**, which makes the prediction task highly challenging. We support this claim with the following evidence:
>
> ### **(a). Energy spectrum analysis**:
>
> In Figure 5(c) in the submission, we could analyze the flow's behavior from the energy spectrum. From the plot, we can see flow contains the following properties:
>
> (a) in an inertial range (wavenumber between 10 and 200), energy is transferred from small scales to large scales.
>
> (b) a dissipation range at high wavenumbers (wavenumber > 200) where energy is removed via viscosity.
>
> In the inertial range, the spectrum maintains a slope of approximately −3, consistent with well-established results in the 2‑D turbulence literature [R1, R2, R3].

---

> ### Author Response · Authors · 2025-08-08
> **P2: Turbulence in Navier-Stokes**
>
> ### **(b). Comparison to Benchmark Studies**:
>
> Flows at similar or lower Reynolds numbers are commonly used as benchmarks in recent scientific machine learning literature, often at much lower grid resolutions.
>
> | Reference                        | Reynolds Number | Grid Resolution      | Notes                |
> | -------------------------------- | --------------- | -------------------- | -------------------- |
> | \[R4] Thermalizer   | 10,000          | 64 $\times$ 64                | Downsampled resolution |
> | \[R5] PDE-Refiner  | 1,000           |  64 $\times$ 64                  | Lower Reynolds number             |
> | \[R6] PIDM         | 1,000           |  256 $\times$ 256                | Lower Reynolds number            |
> | **Ours**                         | 10,000 and 5,000         | **512 $\times$ 512, 256 $\times$ 256** | Higher complexity    |
>
> As seen above, our setup involves a significantly higher resolution and operates at a challenging Reynolds number relative to prior work. This makes both training and inference substantially more difficult.
>
> We hope this provides clarity, and we appreciate your feedback. Thank you.
>
> [R1] Davidson, P. (2015). Turbulence: an introduction for scientists and engineers. Oxford university press.
>
> [R2] Boffetta, G., & Ecke, R. E. (2012). Two-dimensional turbulence. Annual review of fluid mechanics, 44(1), 427-451.
>
> [R3] Tabeling, P. (2002). Two-dimensional turbulence: a physicist's approach. Physics reports, 362(1), 1-62.
>
> [R4] Pedersen, C., Zanna, L., & Bruna, J. (2025). Thermalizer: Stable autoregressive neural emulation of spatiotemporal chaos. ICML 2025.
>
> [R5] Lippe, P., Veeling, B., Perdikaris, P., Turner, R., & Brandstetter, J. (2023). Pde-refiner: Achieving accurate long rollouts with neural pde solvers. NeurIPS 2023
>
> [R6] Shu, D., Li, Z., & Farimani, A. B. (2023). A physics-informed diffusion model for high-fidelity flow field reconstruction. Journal of Computational Physics, 478, 111972.

---

### Official Review · Reviewer_CHmV · 2025-07-01

**Clarity:** 4
**Significance:** 4
**Originality:** 4
**Rating:** 6
**Confidence:** 4

**Summary:**

Authors propose a novel autoregressive multiscale architecture for stable and accurate integration of partial differential equations. Consider a physical field with several resolutions $\phi_h(t), \phi_{2h}(t), \phi_{3h}(t)$ where subscript is a grid spacing. The approach of the authors is, roughly, to predict $\phi_h(t+\Delta t), \phi_{2h}(t + 2\Delta t), \phi_{3h}(t + 3\Delta t)$ from $\phi_h(t), \phi_{2h}(t + \Delta t), \phi_{3h}(t + 2\Delta t)$. This is done by incorporating fields with low resolution into the hidden layers of UNet architecture.

Numerical experiments indicate that implicit neural emulators proposed by authors lead to improved stability, improved accuracy for longer rollouts and better statistical measures (energy spectrum, zonal mean).

**Questions:**

**Main questions:**
1. Suppose we have the scheme with three levels $u\_{n}, z^{(1)}\_{n+1}, z^{(2)}\_{n+2}, z^{(3)}\_{n+3}$. To start iterations we need $u\_0$ (known) and $z^{(1)}\_{1}, z^{(2)\}_{2}, z^{(3)}\_{3}$ (unknown). How coarser state from future time steps are supplied on the first step?

2. A flow of information looks a little peculiar. Why is it important/beneficial to use coarser information from the future to predict the state of the system? For example, to predict $u\_{n+1}$ authors suggest to use not only $z^{(1)}\_{n+1}$ which seems natural but also $z^{(2)}\_{n+2}$, $z^{(3)}\_{n+3}$, etc, which should not be directly related to $u\_{n+1}$. Can the authors please comment on that choice?

3. Related to the previous question. Coarser states are predicted but sometimes not used directly for prediction of corresponding fine state. For example, on step $n-1$, network predicts $u_{n}$ (and also $z^{(1)}\_{n+1}, z^{(2)}\_{n+2}, z^{(3)}\_{n+3}$) using $u\_{n-1}, z^{(1)}\_{n}, z^{(2)}\_{n+1}, z^{(3)}\_{n+2}$ but on previous two time steps $z^{(2)}\_n$ and $z^{(3)}\_n$ were predicted. Presumably, these state are directly related to $u_n$ but not used in its prediction. Did the authors try to evaluate the architecture that uses $u\_{n}, z^{(1)}\_{n+1}, z^{(2)}\_{n+1}, z^{(3)}\_{n+1}$ to predict $u\_{n+1}$ along with coarse latents for future time?

4. Authors show results for $L=1$, $L=2$, $L=3$. Did authors try $L=4$, $L=5$ and so forth? What do authors expect if one keeps increasing the number of coarse levels?

5. Once tacit assumption is that it is easier to predict coarse state for longer lead times. Can the authors support this assumption by theoretical or empirical evidence? If one runs $L=1$ on a coarser version of data is it expected that stability of rollout improves?

6. (Figure 4) Why is the error of the energy spectrum for the $L=3$ case larger than for $L=2$? Simultaneously $L=3$ shows better stability than $L=2$. Similarly, for zonal mean the results seem to indicate that $L=2$ surpasses $L=3$. These results in combination make it slightly unclear whether the increase in the number of levels is beneficial or not. Can the authors please comment on that?

**Minor issues**
1. (line 1, abstract) "Neural PDE solvers offer a powerful tool for modeling complex dynamical systems, ..." I am not sure if modeling is an accurate word in this context. It seems to me solvers are used to approximate a solution when the model is already fixed.

2. (lines 5-8, abstract) "Drawing inspiration from the stability properties of numerical implicit time-stepping methods, our approach leverages predictions several steps ahead in time at increasing compression rates for next-timestep refinements." Sounds ambiguous. Please consider adding "we developed an approach that" after the first comma or replacing "Drawing inspiration from" with "Inspired by".

3. (line 12, abstract) "..., significantly outperforming autoregressive baselines, ..." The method by the authors is autoregressive too, so I would suggest adding the clarification, e.g., "standard autoregressive baselines" or "non-hierarchical autoregressive baselines".

4. (line 102, Section 3) "Consider a nonlinear dynamical system ..." The proposed method uses standard geometrical coarsening so it is suitable for PDEs on spatial domains. All examples the authors provide are for Navier-Stokes equation (equation (13)). I suggest adding to equation (1) some partial derivatives with respect to spatial coordinates. Otherwise it is not clear what is the fundamental difference between (1) and (2) and why one needs a partial derivative in (1). Besides that, line 104 contains an explanation of $\boldsymbol{u}_{t}$ that was never used in (1) and only later appears in (2).

5. (line 212, Section 5) "... (see details in Appendix A.2)" Should be Appendix A.3.

6. (line 255, Section 5.1) It is not explained what the zonal mean is. Please, consider providing reference to the appendix where zonal mean is defined.

**Ethical Concerns:**

["NO or VERY MINOR ethics concerns only"]

**Final Justification:**

My view does not change after rebuttal, so I maintain my original score.

I believe that the authors propose highly original idea that can be useful for many domains. The proposed architecture modifications are supported by thoughtful experimental study with many ablations and comparison with related techniques.

**Limitations:**

yes

**Quality:**

4

**Strengths And Weaknesses:**

**Strengths**
Authors suggest an interesting original research idea and thoroughly confirm it on a set of reasonable experiments. Overall, the principle idea is clearly explained and central claims are well-supported.

**Weaknesses**
Mostly minor typos, unclear parts and several undiscussed issues.

See the section below for details.

---

> ### Author Rebuttal · Authors · 2025-07-31
>
> We greatly appreciate your feedback and your insightful comments on our work! We provide our response as follows:
>
> ### Q1
> > Suppose we have the scheme with three levels $z_{n+1}^{(1)}, z_{n+2}^{(2)}, z_{n+3}^{(3)}$. To start iterations we need $u_0$ (known) and $z_{1}^{(1)}, z_{2}^{(2)}, z_{3}^{(3)}$ (unknown). How coarser state from future time steps are supplied on the first step?
>
> Details on handling the initial cases were briefly provided in Appendix A.3 (see the "Corner Cases" subsection).
> Specifically, during training, with a small probability, we allow the model to receive partially empty latent variables as input and predict the missing ones in the hierarchy.
> For example, in the $L=2$ setting, with a 15% probability, we ask the model to take $[u_n, \mathbf{0}]$ as input and output $z_{n+1}^{(1)}$. During evaluation, we only need a single extra forward run to produce the full latent state $[u_n, \hat{z}_{n+1}^{(1)}]$ to continue the autoregressive rollout.
> Therefore, all methods are evaluated using only a single-frame initial condition for fair comparison. As an example, to generate a sequence trajectory of length $T=1000$, our $L=3$ model performs 1002 forward passes, where the extra $L - 2 = 3$ passes handle these corner cases.
> We will update this discussion in the revised version!
>
> ### Q2
> > A flow of information looks a little peculiar. Why is it important/beneficial to use coarser information from the future to predict the state of the system? For example, to predict $u_{n+1}$ authors suggest to use not only
> $z_{n+1}^{(1)}$ which seems natural but also $z_{n+2}^{(2)}, z_{n+3}^{(3)}$, etc, which should not be directly related to $u_{n+1}$. Can the authors please comment on that choice?
>
> Our key innovation lies in leveraging future information, for example, $z_{n+2}^{(2)}, z_{n+3}^{(3)}$ in $L=3$ method.
> This design can be interpreted in two complementary ways:
> (1) As discussed in Section 4.2, it aligns with the philosophy of multi-step implicit methods. Though not in its canonical form for steps  > 2. As an instance, a rearranged two-step Adams–Moulton $u_{n+1} \approx  u_{n+2} - \Delta t \left(\frac{5}{12} f(u_{n+2}) + \frac{8}{12} f(u_{n+1})-\frac{1}{12}f(u_{n})\right)$ illustrates how future terms contribute to current prediction. And the iterative refinement of $\hat{u}_{n+1}$ through the chain of $\hat{z}_{n+1}^{(1)}$ and  $\hat{z}_{n+1}^{(2)}$ resembles the root-finding algorithms of the implicit solvers.
> (2) From a temporal modeling standpoint, the use of long-range guidance from $z_{n+2}^{(2)}, z_{n+3}^{(3)}$ helps stabilize $\hat{u}_{n+1}$ and improves temporal consistency in long-term forecasts. And with the abstraction strategy of downsampling, our method balances between the model's stability and its capacity to look multi-step ahead.
>
> ### Q3
>
> > Related to the previous question. Coarser states are predicted but sometimes not used directly for prediction of corresponding fine state. For example, on step $n-1$, network predicts $u_n$ (and also $z_{n+1}^{(1)}, z_{n+2}^{(2)}, z_{n+3}^{(3)}$) using $u_{n-1}, z_n^{(1)}, z_{n+1}^{(2)}, z_{n+2}^{(3)}$ but on previous two time steps $z_n^{(2)}$ and $z_n^{(3)}$ were predicted. Presumably, these state are directly related to $u_n$ but not used in its prediction. Did the authors try to evaluate the architecture that uses $u_n, z_{n+1}^{(1)}, z_{n+1}^{(2)}, z_{n+1}^{(3)}$ to predict $u_{n+1}$ along with coarse latents for future time?
>
> Thank you for suggesting this new baseline experiment. Compared to our design, where latent variables $z_{n+1}^{(1)}, z_{n+2}^{(2)}, z_{n+3}^{(3)}$ are distributed across future time steps, the configuration using $z_{n+1}^{(1)}, z_{n+1}^{(2)}, z_{n+1}^{(3)}$ concentrates on all latent variables at the same time step to predict target $u_{n+1}$.
>
> This configuration would likely make the latent states more directly correlated with $u_{n+1}$, potentially simplifying learning. However, we expect that such a design would converge toward would converge into the configuration resembling spatial hierarchy design evaluated in Table 1, with input on $u_n, z_{n+1}^{(1)}, z_{n+1}^{(2)}, z_{n+1}^{(3)}$, where our empirical results find that the hierarchical variables of the same time frame only has minimal improvements compared to the $L=1$ baseline.
>
> ### Q4
> >Authors show results for $L=1, L=2, L=3$. Did authors try $L=4,L=5$ and so forth? What do authors expect if one keeps increasing the number of coarse levels?
>
> Thank you for the suggestion! We conducted additional experiments with
> $𝐿=4$ on the 512×512 resolution turbulent flow dataset. Specifically, we added an extra coarse level $z^{(4)}$ corresponding to an 8×8 resolution, on top of our $L=3$ model. We followed the same training and evaluation protocols as used for $L=3$. The results are summarized below:
>
> | Model | 1 Step | 10th Step | 25th Step | 50th Step |
> |-------|---------|-----------|-----------|-----------|
> | L=3   | 1.41e-03 (3.07e-04) | 1.56e-02 (9.92e-03) | 5.75e-02 (3.04e-02) | **1.81e-01** (9.02e-02) |
> | L=4   | **1.36e-03** (2.47e-04) | **1.40e-02** (4.62e-03) | **5.72e-02** (2.10e-02) | 2.20e-01 (7.93e-02) |
>
> Table 1: Comparison between $L=3$ and $L=4$ for $512\times512$ data.
>
> We find that the $L=4$ model performs slightly better on short-term predictions (≤ 25 steps) but achieves similar results on longer rollouts, when compared to $L=3$. We suspect that the 8×8 resolution may contain too limited information for a $512 \times 512$ input for further significant improvements.
> To conclude, we think that improving the long-term performance of deeper hierarchies may require revisiting the resolution choices of intermediate levels $z^{(1)}, z^{(2)}, z^{(3)}$ to ensure more balanced information flow across scales. And a more complex dataset might lead to more choices and a higher hierarchy.
>
> ### Q5
> >Once tacit assumption is that it is easier to predict coarse state for longer lead times. Can the authors support this assumption by theoretical or empirical evidence? If one runs $L=1$ on a coarser version of data is it expected that stability of rollout improves?
>
> We thank you for suggesting this ablation experiment! Yes! We agree that the ease of predicting coarse states, especially for $L$-step ahead, could partly contribute to our long-term stability.
> One related example was the observation we made in Appendix B.2 regarding a unique recovery pattern seen in our $L=2$ model.
> We will consider adding this ablation study in the final version of the submission.
>
>  ### Q6
> > (Figure 4) Why is the error of the energy spectrum for the $L=3$ case larger than for $L=2$? Simultaneously, $L=3$ shows better stability than $L=2$. Similarly, for the zonal mean, the results seem to indicate that $L=2$ surpasses $L=3$. These results, in combination, make it slightly unclear whether the increase in the number of levels is beneficial or not. Can the authors please comment on that?
>
> Figure 4 primarily benchmarks long-term rollout stability. Since the MSE metric is less informative in this setting, we instead assess the model's ability to preserve physical properties. Given the extended rollout horizon, the variance is significant (see the shaded regions in Figure 4(a)). Both $L=2$ and $L=3$ models show good alignment with the ground truth and perform substantially better than the baseline methods. In this figure, the comparison between $L=2$ and $L=3$ is less critical than their overall improvement over the baselines. In contrast, Figure 3(b) offers a more stringent criterion for long-term evaluation, under which the performance gap between $L=2$ and $L=3$ becomes more pronounced.

---

> ### Comment · Reviewer_CHmV · 2025-08-08
>
> I would like to thank the authors for detailed reply, and especially for conducting additional experiments. I keep my original score.
>
> In my view the idea proposed by authors is exceptionally interesting, especially in the context of temporal abstractions for prediction and control.
>
> Other reviewers raised concerns that the proposed approach is not general enough since vanilla U-Net architecture is restricted to uniform discretization. I believe it is straightforward, if tedious, to define and train U-Net-like architectures on unstructured grids [1], [2], [3].
>
> [1] Graph coarsening: from scientific computing to machine learning l. J Chen, Y Saad, Z Zhang
>
> [2] Algebraic multigrid JW Ruge, K Stüben
>
> [3] Multipole graph neural operator for parametric partial differential equations. Z Li, et al

---

### Note · Authors · 2025-08-13

We thank all reviewers and the area chair. Specifically, we thank reviewer **CHmV** for **acknowledging our work and providing invaluable feedback**! We thank reviewer **PLCM** for **acknowledging that our rebuttal addressed the raised concerns** and agreeing to update the rating!

We thank reviewers **HWeY** and **vohk** for invaluable discussions, and we **sincerely hope that our latest results on comparison methods will support a holistic evaluation of our work for your consideration**.

Below, we summarize our primary technical contributions to facilitate a holistic evaluation.

## Methodology:
We propose a novel autoregressive model, inspired by implicit numerical schemes, that leverages a hierarchy of representations from both past and **future** states to capture multiscale interactions in complex dynamical systems. This design delivers substantial gains in (1) short-term accuracy and (2) extremely long-term stable forecasts, while scaling efficiently across multiple hierarchies with minimal computational overhead.

## Results:
We evaluate our method on a highly challenging setting: chaotic turbulence at Reynolds number Re = $10^4$ and and high spatial resolutions ( 256$\times$256 and 512$\times$512) using the Navier–Stokes dataset. Key results are summarized below.

**Table: Short-term ($\leq$ 100 rollout steps) and long-term ($2 \times 10^5$ rollout steps) performance**


|Method|Seconds per training iter.| 1-step MSE | 25-step MSE | 50-step MSE |$[0, 2\times10^5]$ steps zonal mean error | $[0, 2\times 10^5]$ steps energy spectrum error|
|:--:|:--:|:--:|:--:|:--:|:--:|:--:|
|Unrolled training[r1]|0.419 (0.003) |5.94e-04 (1.32e-04) | 9.96e-02 (5.04e-02) | 4.06e-01 (1.81e-01) | 3.54 (1.9)|3.07e+01(6.03e-01)|
|Pushforward[r2]|0.283 (0.003) |1.08e-03 (2.26e-04)	|9.16e-02 (4.62e-02)|3.19e-01 (1.34e-01) | 1.07 (0.44)|2.41e+01(1.06e+00)|
|Baseline: L=1|0.207 (0.002) |5.60e-04 (1.15e-04)	|8.04e-02 (3.45e-02) | 3.12e-01 (1.15e-01) |4.20 (2.36)|2.48e+02 (8.93e+00)|
|Ours : L=3|0.220 (0.002) |**5.50e-04** (1.20e-04)	|**3.37e-02** (1.41e-02)| **1.40e-01** (6.16e-02)  |**0.63** (0.58)|**3.91e+00** (1.80e-01)

[r1] Differentiability in unrolled training of neural physics simulators on transient dynamics, Bjoern List, Li-Wei Chen, Kartik Bali, Nils Thuerey,

[r2] Message passing neural pde solvers, Johannes Brandstetter, Daniel Worrall, and Max Welling

---

### Decision · Program_Chairs · 2025-09-17

**Decision:**

Accept (poster)

**Comment:**

The paper introduces a contribution to the field of neural PDE solvers by addressing two challenges: long-term stability and error accumulation.  The proposed hierarchical implicit neural emulator is original and effective.

While some reviewers state that theoretical aspects could have been explored, they also found that the current empirical evidence is robust and convincing. Rebuttals have been key in the discussion and we are happy to propose to accept the paper.